# Atmospheric influences on the anomalous 2016 Antarctic sea ice decay

**Elisabeth Schlosser[1,2], F. Alexander Haumann[3,4], Marilyn N. Raphael[5]**

[1]Institute of Atmospheric and Cryospheric Sciences, University of Innsbruck, Innsbruck, Austria}

[2]Austrian Polar Research Institute, Vienna, Austria}

[3]Environmental Physics, Institute of Biogeochemistry and Pollutant Dynamics, ETH Zürich, Zürich, Switzerland}

[4]British Antarctic Survey, Cambridge, UK}

[5]Department of Geography, University of California, Los Angeles, California, USA}

*Correspondence to:* E. Schlosser (Elisabeth.Schlosser@uibk.ac.at)

submitted to: The Cryosphere, 1 September, 2017

revised version: 5 January, 2018

final version: 2 March 2018

**Abstract** In contrast to the Arctic, where total sea ice extent (SIE) has been decreasing for the last three decades, Antarctic SIE has shown a small, but significant increase during the same time period. However, in 2016, an unusually early onset of the melt season was observed; the maximum Antarctic SIE was already reached as early as August rather than end of September, and was followed by a rapid decrease. The decay was particularly strong in November where Antarctic SIE exhibited a negative anomaly (compared to the 1979–2015 average) of approximately 2 million $km^2$. ECMWF-Interim reanalysis data showed that the early onset of the melt and the rapid decrease in sea ice area (SIA) and SIE were associated with atmospheric flow patterns related to a positive zonal wave number three (ZW3) index, i.e. synoptic situations leading to strong meridional flow and anomalously strong southward heat advection in the regions of strongest sea ice decline. A persistently positive ZW3 index from May to August suggests that SIE decrease was preconditioned by SIA decrease. Particularly, in the first third of November northerly flow conditions in the Weddell Sea and the Western Pacific triggered accelerated sea ice decay, which was continued in the following weeks due to positive feed-

back effects, leading to the unusually low November SIE. In 2016, the monthly mean Southern Annular Mode (SAM) index reached its second lowest November value since the beginning of the satellite observations. A better spatial and temporal coverage of reliable ice thickness data is needed to assess the change in ice mass rather than ice area.

## 1    Introduction and previous work

Sea ice plays a critically important role in the cryosphere as well as the global climate system. It modifies heat and mass exchange processes. Sea ice has a very high albedo compared to open water, thus when present, it greatly increases the surface reflectivity. It also reduces the exchange of heat and moisture between the ocean and the lower atmosphere. Strong positive feedback effects related to sea ice changes have consequences for both atmosphere and ocean. Melting of sea ice rapidly decreases the surface albedo and increases turbulent heat exchange between ocean and atmosphere; it thus strongly influences the surface energy balance of the ocean (e.g. Stammerjohn et al., 2012), which in turn affects sea ice itself. Formation and melt of sea ice also influences the sea water salinity by releasing salt (brine rejection) upon freezing and fresh water upon melting. These fluxes dominate the surface freshwater flux balance of the seasonally sea ice covered Southern Ocean and thereby strongly influence the ocean overturning circulation (Haumann et al., 2016), which is critical for exchanging heat and carbon between the deep ocean and the atmosphere. Therefore, Antarctic sea ice plays an active role in the global climate system.

While in the Arctic the total sea ice extent (SIE) has been decreasing for more than three decades (e.g. Simmonds, 2015), Antarctic SIE has shown a small, but significant increase during the same time period (1.5 % per decade over the period 1979 to 2013; Turner et al., 2015), reaching record extents in 2012, 2013, and 2014 (Massonnet et al., 2015; Reid and Massom, 2015). However, in 2016, an unusually early onset of the seasonal ice melt was observed. The maximum Antarctic SIE was reached a month earlier, in August, rather than end of September, and was followed by rapid decrease until the summer when record minimum SIE was observed. The sea ice area (SIA), the total area actually covered by ice, started to exhibit values below the long-term average as early as July, with only a brief return to normal in late August when the general sea ice retreat began. The decay was particularly strong in November, when Antarctic SIE exhibited a negative anomaly (compared to the 1979–2015 average) of 1.84 million $km^2$, which is unprecedented in the satellite era (since 1979) and, combined with reduced Arctic sea ice, led to a distinct minimum in global SIE (NSIDC, 2016).

The present study is motivated by this unusual behavior of Antarctic sea ice in 2016. Sea ice retreat is influenced by atmospheric as well as oceanic processes. Local atmospheric influences on sea ice, both dynamic and thermodynamic, act on a relatively short time scale with an immediate response of the

sea ice. In this study, we solely focus on the local atmospheric influences on sea ice retreat and discuss related possible reasons for the observed anomalous sea ice decay.

Locally, changes in SIE and SIA can be caused by melting or freezing and by sea ice dynamics that
influence ice advection, ice divergence and convergence (e.g. Enomoto and Ohmura, 1990). On a short time scale (days to weeks), both processes are strongly dependent on atmospheric influences, i.e. thermodynamics of the ocean–atmosphere interface and wind stress on the floating ice (e.g. Watkins and Simmonds, 2000). Oceanic influences mainly play a role on longer time scales, such as seasonal and interannual variations (e.g. Gordon and Huber, 1990; Martinson, 1990). In particular, while
changes in the timing of sea ice retreat seem to have a strong influence on the timing of the following advance, advance and subsequent retreat are only weakly related (Stammerjohn et al., 2012). This relation suggests a strong oceanic thermal feedback that accelerates or decelerates autumn freeze onset (Stammerjohn et al., 2012). In a recent study, Su (2017) stated that the location of the sea ice edge at the SIE maximum is strongly correlated with the upper-ocean stratification in early winter (April–
June), consistent with the findings by Lecomte et al. (2017). The seasonal influence of the ocean on SIE has been the subject of a number of studies. Hall and Visbeck (2002) show that the strength and position of the westerlies influence northward Ekman transport and therefore the SIE as well as upwelling of warmer water closer to the continent. This effect can cause stronger retreat of sea ice in summer, when the ice edge is farther south, and expansion of sea ice in winter (e.g. Lee et al., 2017).

Enomoto and Ohmura (1990) investigated the relationship of the sea ice edge advance and retreat and the semi-annual oscillation of the circumpolar trough. The ice drift depends on the position of the sea ice edge relative to the Antarctic convergence line (Enomoto and Ohmura, 1990), which moves according to the semi-annual oscillation of the circumpolar trough (Meehl, 1991). This leads to ice divergence when the ice edge is north of the convergence and transport of ice towards the coast, when
the ice edge is south of the Antarctic convergence line. The velocity of the ice drift amounts to 1–2 % of the surface wind speed; the direction of the ice drift is turned by 10–40° to the left of the direction of the atmospheric flow, an average value of approximately 30° was found (Kottmeier et al., 1992; Kottmeier and Sellmann, 1996; Nansen, 1902). Therefore, the meridional ice transport that affects the latitudinal extent of the sea ice is primarily driven by meridional winds, but also zonal winds can have
an influence, particularly in the Ross and Weddell Seas where the sea ice edge extends to lower latitudes into the westerly wind belt during winter (Haumann, 2011). Holland and Kwok (2012) investigated the influence of surface winds on Antarctic sea ice changes over the period 1992 to 2010. They state that while in most parts of West Antarctica mainly wind-driven changes in ice advection are responsible for the observed sea ice concentration (SIC) trends, wind-driven thermodynamic changes
dominate in all other parts of the Southern Ocean. Haumann et al. (2014) and Kwok et al. (2017) show that meridional wind changes also play a major role in driving the sea ice trends over the entire satellite record. In some regions, zonal winds are also important for zonal redistribution of the ice

(Haumann, 2011; Kwok et al., 2017). This means that both changes in the meridional and zonal components of the atmospheric circulation can induce sea ice anomalies, either through associated temperature anomalies or through ice transport anomalies.

Raphael (2007) studied the influence of atmospheric zonal wave number three (ZW3) on Antarctic sea ice variability. ZW3 (together with ZW1) describes the asymmetry in the generally strongly zonally symmetric circulation in the extratropical Southern Hemisphere. It is related to a pattern of alternating northward and southward flow, which influences the temperature difference between ocean and atmosphere and thus their exchange of sensible heat. Raphael (2007) states that sea ice responds to these heat fluxes. Pezza et al. (2012) investigated the relationship between recent SIE extremes and the Southern Annular Mode (SAM), the dominant mode of atmospheric variability in the extra-tropical Southern Hemisphere (Marshall, 2003), as well as El Niño-Southern Oscillation (ENSO). They found that the relationship is not uniform for all longitudes and that the natural variability of SIE and the SAM is so large that a longer reliable time series of sea ice and climate data would be necessary to establish a causal relationship between the two variables. Simpkins et al. (2012) show that the relation of SAM and ENSO with Antarctic sea ice anomalies exhibits a strong regional and seasonal dependence through the associated atmospheric circulation patterns. Simulations with global climate models indicate that also the oceanic response to SAM leads to interannual variations of Antarctic sea ice, with an initial expansion of the sea ice due to positive SAM anomalies during austral summer (Holland et al., 2017). Moreover, modeling studies suggest a strong influence of interannual and decadal tropical variability, especially in the Pacific, on Antarctic sea ice anomalies through the atmospheric circulation (Meehl et al., 2016; Purich et al., 2016; Schneider and Deser, 2017) and oceanic feedbacks (Stuecker et al., 2017).

In the past 37 years covered by satellite data, variations in the total Antarctic SIE have been small since large contrasting regional variations, both positive and negative deviations from the mean, cancel each other. Large regional anomalies in SIC can be caused by stable synoptic situations, where atmospheric flow conditions that persist over several days or are dominant over several weeks lead to a southward (northward) ice transport combined with warm (cold) air advection (e.g. Massom et al., 2006; Schlosser et al., 2011). Turner et al. (2009) stated that the long-term increase in Antarctic SIE is mostly caused by an increase in the Ross Sea sector, which is strongly influenced by the strength and position of the Amundsen Sea Low (ASL) (Raphael et al., 2015). Turner et al. (2009) also related stratospheric ozone loss to the observed increase in Antarctic SIE and found that the strengthening of the westerlies, which is related to ozone depletion, deepens the ASL. They conclude, however, that the observed SIE increase might be still within the limits of natural variability. Haumann et al. (2014) found that, on multidecadal time scales, changes in Antarctic SIE are linked to changes in the strength of meridional winds that are caused by a zonally asymmetric decrease in high-latitude surface pressure, the latter possibly being related to stratospheric ozone depletion, greenhouse gas increase, or

natural variability. While increased southerly (and potentially westerly) winds most likely fueled the long-term increase of Antarctic sea ice over the satellite era, we here hypothesize that increased northerly winds and an associated warm air advection triggered the negative Antarctic sea ice anomaly in 2016.

Two recent studies investigated the observed unusually fast retreat in the spring of 2016. Turner et al. (2017), after a discussion of the climatological mean behavior of Antarctic sea ice, consider the spatial differences of SIE anomalies and their temporal changes for five different sectors of the Southern Ocean. While in the Weddell Sea a marked change from positive to negative SIE anomalies occurred in November, the Amundsen-Bellingshausen Seas and the Indian Ocean showed mainly negative anomalies throughout the melt season. The Ross Sea and West Pacific sectors exhibited both positive and negative SIE anomalies during the entire period. They argue that the decrease of SIE at a record rate is linked to record surface pressure anomalies, mainly in the Weddell Sea and Ross Sea sectors of Antarctica, and the related meridional flow and corresponding warm air advection (Turner et al., 2017). Alternatively, Stuecker et al. (2017) hypothesize that the anomalously low SIE in November–December 2016 was largely related to positive sea surface temperature anomalies due to the extreme El Niño event peaking in December–February 2015/16 that persisted throughout the winter and the concurrent negative phase of the SAM.

Generally, changes in both oceanic and atmospheric processes could have been responsible for the observed 2016 anomaly. In the present study we investigated in detail the onset and temporal evolution of the sea ice retreat (considering both SIA and SIE, as well as SIC) and the contributions of various parts of the Southern Ocean. A detailed analysis of the evolution of the SIA and SIE deficits shows a rapid growth of regional anomalies, suggesting an atmospheric origin of these changes. We compared those changes to ice drift data and related it to the general atmospheric flow conditions derived from the European Centre for Medium-Range Weather Forecasts (ECMWF) reanalysis data and considered the relationship to SAM and ZW3 indices and patterns.

## 2    Data and methods

### 2.1    Sea ice data

Sea ice data are provided by the National Snow and Ice Data Center (NSIDC). SIC is derived from Nimbus-7 SMMR and DMSP SSM/I-SSMIS passive microwave data (Maslanik and Stroeve, 1999). The data set is generated from brightness temperature and designed to provide a consistent time series of SIC. The data are provided in polar stereographic projection at a grid cell size of 25 km x 25 km. We use the Climate Data Record version 2 NASA Team algorithm (CDR NT; Meier et al., 2013, updated 2016) data for the period 1979 to 2015 as a baseline climatological record. These data are

compared to the Near-Real-Time data (NRT; http://nsidc.org/data/NSIDC-0081; Maslanik and Stroeve, 1999) from January 2015 to June 2017, which are also derived from passive microwave data using the NASA Team algorithm. Days with a large number of missing pixels in the SIC have been removed from the NRT data. The resulting SIE anomalies for the overlapping year 2015 agree well (Fig. 1). To create monthly mean ice drift fields the low-resolution sea ice drift product of the EUMETSAT Ocean and Sea Ice Satellite Application Facility (OSI SAF, www.osi-saf.org; Lavergne et al., 2010) were used.

SIE is defined as the spatial sum of the area of grid boxes that have a SIC of at least 15 %. SIA is defined as the spatial sum of the SIC times the grid box area for grid boxes that have a SIC of at least 15 %. The daily SIA change was computed by first calculating the daily SIA and then taking the difference in SIA from one day to the next. We computed daily and monthly anomalies as well as SIE and SIA deficits with respect to the climatological daily and monthly means of the CDR NT record over the period 1979 to 2015. Long-term trends shown in Fig. 1 are computed using linear regression analysis.

## 2.2    ECMWF reanalysis data

ECMWF-Interim reanalysis (ERA-Interim) data were employed to investigate atmospheric flow conditions. The reanalysis data are available from 1979 to present. The horizontal resolution is T255, corresponding to approximately 79 km. The model has 60 vertical levels, with the highest level being at 0.1 hPa. A detailed description of ERA-Interim is given in Dee et al. (2011). We used monthly mean sea level pressure and vertically integrated northward heat flux, derived from the daily mean values. The reanalysis data were used for investigation of ZW3 patterns and analysis of surface pressure fields and cyclone activity.

## 2.3 SAM and ZW3 indices

Meridional transport of heat and moisture in the high latitudes of the Southern Hemisphere is carried out by the asymmetric component of the circulation also known as quasi-stationary waves (Raphael, 2004). Together, zonal wave one (ZW1) and ZW3 represent the largest proportion of this asymmetry (van Loon and Jenne, 1972). The asymmetry described by these zonal waves is revealed when the zonal mean is subtracted from the geopotential height field and coherent pattern of zonal anomalies and their associated flow become apparent. Both ZW1 and ZW3 have preferred regions of meridional flow, guiding the meridional transport of heat and moisture into and out of the Antarctic. ZW1 and ZW3 vary with each other so that when ZW3 is strong, ZW1 is weak and vice versa. An index of ZW3 based on its amplitude defined by Raphael (2004) shows that ZW3 has positive and negative phases

where a positive phase indicates strong meridional flow and a negative phase more zonal flow. A climatology of this index shows that ZW3 tends to be negative from September through December, a period of time when ZW1 is dominant in the field (Raphael, 2004). Since it exerts control on the meridional flow, ZW3 influences the variability of Antarctic sea ice dynamically and thermodynamically as illustrated in Raphael (2007).

The SAM is the dominant mode of atmospheric variability in the extra-tropical Southern Hemisphere. Marshall (2003) calculated a SAM index based on the observed pressure difference between stations in Antarctica and at mid-latitudes. A large (small) meridional pressure gradient means a positive (negative) SAM index and, correspondingly strong (weak) westerlies. The stronger westerlies are usually connected to a more zonal flow, whereas in the case of a negative SAM index, the flow tends to meander in amplified Rossby waves. A positive SAM index is mostly related to positive SIC anomalies (Simpkins et al., 2012) since, apart from the Antarctic Peninsula, air temperatures are lower due to decreased meridional heat exchange and the stronger westerlies can lead to increased northeastward transport of sea ice. A northerly flow during periods of negative SAM index can lead to compaction of the ice and/or melting due to warm air advection. The corresponding southerly flow means offshore flow, cold air advection and thus new ice formation. A negative SAM thus usually is associated with large regional differences in SIC and SIE anomalies. However, the SAM explains only approximately one third of the variability (Marshall, 2007) and there can be large regional differences for both the positive and the negative mode of SAM, e.g. strong zonal flow in the Pacific and strong meandering in the western Atlantic at the same time. In the present study we use the ZW3 and SAM indices to help explain the anomalous retreat of Antarctic sea ice in 2016.

## 3  Results

### 3.1 Temporal evolution of Antarctic sea ice extent and area

In Fig. 1 monthly SIE anomalies from January 1979 to June 2017 are displayed. Until 2015, only relatively small variations and a generally positive trend can be seen. This trend is abruptly interrupted in November 2016, when monthly mean SIE was almost 2 million $km^2$ lower than in the preceding year. This value is clearly outstanding: the negative deviation from the mean reaches almost four times the standard deviation (2.06 million $km^2$) of the 1979 to 2015 period. Trends are calculated from the CDR data until the end of 2015 and for the period including the most recent 17 months from NRT data, respectively. The extension of the period by these latter 17 months reduces the positive trend from 0.24 million $km^2$ per decade (1979–2015) to 0.16 million $km^2$ per decade.

Figure 2 shows the daily Antarctic SIE (Fig. 2a) and SIA (Fig. 2b) for the year 2016 together with the 1979–2015 average. The gray shading refers to the two standard deviations (1979–2015). While, on average, SIE increases until the end of September, in 2016, a sudden decrease occurred in the last third

of August. After the first week of September SIE fell below the long-term average and never recovered during the rest of the year. The monthly mean anomaly in SIE increased from $-0.1$ million km$^2$ in August, to a maximum deviation from the monthly mean of $-1.84$ million km$^2$ in November. In December the anomaly amounted to $-2.33$ million km$^2$. The SIA exhibited negative deviations from the mean already in June and July, briefly reached values close to the average again at the end of August and thereafter remained below the average for the rest of the year. The monthly mean SIA anomaly for November and December 2016 are $-1.98$ million km$^2$ and $-1.55$ million km$^2$, respectively, and generally preceded the SIE anomalies. The SIE deficit (shaded pink vs. dashed grey two standard deviations (1979–2015)) also suggests that there were two distinct periods of decline, the first one in late August to mid-September, and the second one in the first half of November. While the first period saw a corresponding decline in SIA, the second period is not as distinct for SIA. The overall largest SIA deficit of 2.24 million km$^2$ occurred on 18 November, whereas the largest SIE deficit of 2.8 million km$^2$ was seen approximately one month later, on 14 December. The sea ice never did recover in austral summer 2016/2017; in February 2017 the monthly mean SIE amounted to 2.27 million km$^2$, which is the lowest February value since beginning of the satellite measurements in 1979. However, the monthly mean SIA was with 1.63 million km$^2$ only the second lowest one, here the absolute minimum occurred in February 1993 (1.4 million km$^2$).

In Fig. 2c the SIA change (daily temporal derivative of the SIA) is shown. Again, the red and black lines refer to 2016 and the climatological mean (1979–2015), respectively. The SIA change can be interpreted as ice melt or compaction when it is negative and as new ice formation when it is positive. The pink shading indicates anomalously high melt or low freezing periods, and the blue shading the opposite. A first, very short period with SIA decrease was observed as early as in mid-July when the SIA change reached negative values for the first time.

It can be clearly seen that, until August, periods with growing and shrinking SIA deficit alternate, whereas from end of August on, almost only negative deviations from the mean are found, with negative SIA changes for the rest of the year. In addition to the periods mentioned above, a further period with strong sea ice decline can be seen in mid-October.

## 3.2 Regional sea ice changes

### 3.2.1 SIC anomalies and corresponding influence of cyclonic activity

To investigate which areas contributed predominantly to the observed changes, Antarctica was divided into seven sub-areas (see Table 1), the definition following specifically the regional SIC anomalies displayed in Fig. 3a–d for the months August to November 2016. This division is finer than the five sub-areas used in previous studies (e.g. Turner et al., 2017) and thus allows a detailed discussion of the temporal and spatial development of the sea ice retreat. In particular, the sub-division of the Weddell

Sea was necessary due to the different behavior of the western and eastern part. The SIC anomalies, as usual, exhibit strong regional differences, areas with positive deviations alternating with areas of negative deviations. In August, the total SIE is only 0.1 million km$^2$ smaller than the long-term average (1979–2015). The contributions of areas with negative SIC anomalies mostly counteract those of areas with positive anomalies. Largest negative SIC anomalies are found at the eastern edge of the Eastern Weddell Sea (EWS) and in the Western Indian Ocean (WIO), where also the sea ice edge (grey contour line in Fig. 3a) is already distinctly farther south than on average (green contour line). In the Eastern Indian Ocean (EIO), only a very small area, centered at approximately 125° E shows negative anomalies of SIC, otherwise SIC anomalies are slightly positive. In the Western Pacific / Western Ross Sea (WP), the sea ice edge is also close to the long-term average; positive deviations in SIC are found near 150° E and at the northern edge of the Ross Sea. The Amundsen-Bellingshausen Seas (ABS) show southward deviations from the mean sea ice edge, whereas both the Western Weddell Sea (WWS) and EWS exhibit positive SIC anomalies between approximately 35° W and 0°, associated with a sea ice edge farther north than on average.

Generally, SIC depends on ice dynamics (advection) and thermodynamics, both, on short time scales, being strongly influenced by the atmospheric flow. Surface winds above the Southern Ocean depend mostly on the cyclone activity in the circumpolar trough (Simmonds et al., 2003). Figure 3i–l shows the monthly mean surface pressure for the corresponding months. Arrows indicate the approximate main surface flow direction related to changes in SIC or sea ice edge. In the Southern Hemisphere, the air circulates clockwise around the center of a cyclone, meaning a northerly (southerly) flow at the western (eastern) flank of the cyclone. Northerly flow leads to compaction of the ice and/or advection of warmer air, and thus a southward displacement of the sea ice edge by compaction and/or melt. In contrast, a southerly flow is related to northward drift of the ice and formation of new ice close to the coast due to cold air advection from the continent over leads and polynyas that develop as the ice is pushed away. The strong westerly flow at the northern edge of the cyclone can, via Ekman transport (e.g. Hall and Visbeck, 2002), also lead to net northward transport of sea ice and thus a northward displacement of the sea ice edge. In Fig. 3e–h the vertically integrated northward heat flux from ERA-Interim is displayed. Positive (negative) values suggest northward transport of cold (warm) air, which in most cases in Antarctica means cold (warm) air advection. Strictly spoken, since advection is proportional to the meridional temperature gradient, the heat flux alone does not necessarily prove warm air advection. However, the spatial temperature distribution around Antarctica is basically zonal, thus a northerly (southerly) flow is practically always associated with warm (cold) air advection. It has also been shown in Antarctic precipitation studies (e.g. Gorodetskaya et al., 2014) that considerable warm air (and thus moisture) advection in the circumpolar trough is usually observed in a thick layer from the surface to the upper troposphere, which influences not only the sensible heat flux at, but also longwave radiation to the surface, both favoring sea ice melt. A strong signal in the vertically integrated meridional heat flux thus supports our findings. The air mass transport agrees very well

with the general flow direction derived from the surface pressure field (see Fig. 3i–l and arrows herein).

In August (Fig. 3i), the Amundsen Sea Low (ASL) was replaced by an extended cyclone centered above the eastern Ross Sea, causing strong southerly flow at its western flank and positive SIC in the WP. Correspondingly, the northwesterly flow at its eastern flank of the ASL causes negative SIC anomalies in the Eastern Ross Sea (ERS) and in the ABS. The small, but distinct negative SIC anomaly around 125° E is related to a low pressure system in the EIO/WP that steered relatively warm air masses from areas north of 55° S towards the mentioned anomaly (Fig. 3e). The area is rather restricted due to the confluent flow. In the EWS and the WIO, there was a weak, but extended, low pressure system that caused a weak northerly flow and warm air advection towards the western edge of the WIO associated with the negative SIC anomalies. However, most of the WIO was under the influence of a very weak pressure gradient, and consequently no distinct flow conditions prevailed.

In September (Fig. 3b), positive SIC anomalies occurred in the EWS, EIO, and ERS. The area with negative anomalies around 125° E extended eastward across the entire WP. Additionally, SIC anomalies in the ABS and WWS were largely negative. The low pressure system in the Pacific Ocean (Fig. 3j) intensified and moved eastwards, extended across the Ross and Bellingshausen Seas, which resulted in positive (negative) SIC anomalies at its western (eastern) flank. The WP was still influenced by the warm air advection on the low centered in the EIO. The positive anomalies in the EIO are associated with an area of cold air advection (Fig. 3f and j). The negative SIC anomalies between 180° W and approximately 160° W (WP) are unexpected since this area is influenced by the ASL. In the northern parts of the EWS, positive SIC anomalies are observed, possibly related to the strong westerly flow resulting in zonal and meridional ice advection (Fig. 6b) due to a northeastward Ekman transport.

In October (Fig. 3c), the SIC anomalies occurred in the same areas as in September, but were more strongly developed. The ASL stretched eastward (Fig. 3k) and extends now across the ABS. In the EWS, a strong low pressure system has developed, leading to a northerly flow and warm air advection in the WIO, where strong negative SIC anomalies are observed. The Eastern Indian Ocean/Western Pacific low extended northwards, although with a weaker pressure gradient. SIE in the EIO (Fig. 5g) changed very little and slightly less negative SIC anomalies occurred. The negative anomaly in SIE of the WIO (Fig. 5d) remained nearly constant in September and October.

In November, the WIO exhibited strong negative anomalies in SIC (Fig. 3d) and SIE (Fig. 5d), which contributed to the pronounced melt period in the first half of November (Fig. 2c). Also large negative deviations from the mean SIC (Fig. 3d) occurred in the WP, the ABS, and in the WWS, whereas positive deviations in SIC were found around the sea ice edge in the EWS, and the ERS. In the southeastern Weddell Sea (Fig. 3d), negative SIC anomalies prevail, and the opening of the Weddell

Sea polynya (see also section 3.2.3), which reoccurred more strongly in 2017 (not shown), over Maud
Rise becomes apparent.

Since we are considering monthly means here, a signal that is still visible in a monthly mean has to
have been predominant over the majority of days during that month or has to have been very strong
over a shorter time period. The latter was the case at the end of August, and also in November: in the
first third of the month the prevailing northerly flow with warm air advection was very strong and thus
crucial for triggering the increased melt. This situation is shown in Fig. 4, where the meridional heat
flux (Fig. 4a) and the mean sea level pressure (Fig. 4b) for 1–10 November 2016 is displayed. It
suggests that the strongest contribution to the monthly mean meridional heat flux (Fig. 3h) stems from
the first third of the month. Figure 4b shows the corresponding surface pressure.

Between an extended low pressure system above the Eastern Indian Ocean/Western Pacific and a high
pressure ridge in the Ross Sea, a strong northerly flow with warm air advection induced large negative
SIC anomalies in that area. Similarly, in the western and central Weddell Sea, a northeasterly flow at
the eastern flank of a low pressure system is associated with strong negative SIC anomalies. In the
WIO, the picture is not as straightforward; oceanic memory effects may play a role here.

### 3.2.2 Temporal evolution of regional SIE and SIA

To get a deeper insight into the spatial origin of the anomalies in SIE and SIA, in Fig. 5 the daily
values of SIE, SIA, and SIA changes are displayed (similar to Fig. 2, for details see figure caption) for
the different regions shown in Fig. 3. The EIO and ERS regions did not contribute to the accelerated
sea ice decay. They have predominantly positive anomalies in SIE, SIA and SIC. The EWS, WP, and
WWS regions contributed to the negative SIE and SIA anomalies through strong events associated
with the corresponding atmospheric flow discussed above, whereas the WIO and ABS exhibited more
or less continuous negative deviations from the long-term mean. These developments are most clearly
seen in the respective deficits of SIE and SIA shown at the bottom of the Fig. 5 plots. In the EWS,
both SIA and SIE stayed above the long-term average, but contributed strongly to the negative
anomaly through a distinct event evolving in November.

The first slightly negative SIA changes in mid-July occurred in the WP (Fig. 5l), the ABS (Fig. 5r),
and the WWS (Fig. 5u). The start of the melt period at the end of August is also seen in these sectors
(except for the EWS). The WIO exhibited negative SIE (Fig. 5d) and SIC anomalies in August (Fig.
3a), even though the SIA change (Fig. 5f) hinted at anomalously high freezing in this area at the end of
August. The negative anomalies in this area might be an oceanic memory effect since the area showed
similar anomalies in the preceding months (Fig. 5d–e) and even in the preceding year (Stuecker et al.,
2017). The SIA change in the two regions WP (Fig. 5l) and ABS (Fig. 5r) also contributed to the

continuation of the late August melting into the first half of September. The EWS did not contribute significantly to the sea ice decay in these early periods. In September, the contribution of the WWS increased considerably, whereas the EWS (Fig. 5c) contributed late, but strongly from mid-November into December to the negative sea ice anomalies.

### 3.2.3 Ice drift

Ice drift influences SIE, SIA and SIC in various ways. For example, northerly winds close to the ice edge lead to a southward sea ice compaction when the SIC is well below 100 %, and ridging and rafting, i.e. ice thickening, when the SIC is close to 100 %. Southerly winds close to the coast imply offshore transports that can cause coastal polynyas. These polynyas are usually quickly closed by new ice formation due to associated cold air advection from the continent. Strong westerly winds can lead to an eastward redistribution of the ice with a northerly component due to Ekman transport. Apart from that, convergence/confluence and divergence/diffluence in the main atmospheric flow can locally change SIC.

Since sea ice drift is strongly influenced by surface winds and might have contributed to the 2016 SIC anomalies, we examine the monthly mean ice motion from August to November 2016, illustrated in Fig. 6. It shows generally good agreement with the sea level pressure fields shown in Fig. 3, taking into consideration the corresponding wind fields discussed in section 3.2.1. Only in areas where SIE is already small, i.e. in the EIO and the eastern half of the ABS in October and November, the correlation between ice drift and wind direction more or less disappears due to the low resolution of the plotted ice drift. In August, the large low pressure system above the Weddell and Lazarev Seas is associated with clockwise ice drift in the Weddell Sea and drift towards the continent at its eastern flank. With an increasing pressure gradient in September, the drift increases at the northern flank, whereas the southward drift at its eastern flank disappears due to a predominantly westerly flow in the Lazarev Sea. The ice drift connected to the ASL is best developed in September, with strong southerly (northwesterly) flow at the western (eastern) flank of the cyclone, contributing to the positive SIC anomaly in the ERS (Fig. 3b).

In November, the monthly mean sea ice drift direction and strength (Fig. 6d) are strongly connected to the average sea level pressure field, which is largely induced by anomalies during the first third of the month (1–10 November 2016, shown in Fig. 4b). The patterns of the strong clockwise circulation in the Western Weddell Sea lead to a southward compaction and thus a quickly decreasing SIC around the northern ice edge region. The divergent atmospheric flow over the southeastern Weddell Sea in August and November (Fig. 3i and l) leads to a strongly divergent sea ice drift in southeastern Weddell Sea, especially in November (Fig. 6d), which potentially contributed to the first reoccurrence of the Weddell Sea polynya since the 1970s (Mazloff et al., 2017; Reid et al., 2017). The dynamical removal

of the sea ice from this region just before the ice melt, when the ocean surface density stratification is weak, can trigger strong positive feedbacks (Gordon, 1991) that might have led to an even more

pronounced reappearance of the Weddell Sea polynya in 2017 (not shown). The drift towards the coast in the WP (negative SIC anomaly in Fig. 3d) and the offshore drift in the Eastern Ross Sea (positive SIC anomaly in the ERS in Fig. 3d) in November are contributing to the SIC anomalies, and can be clearly related to the pressure field and the corresponding surface winds.

**3.3 General atmospheric flow conditions during the melt season 2016**

After having investigated in section 3.2.1 the local meridional atmospheric flow related to sea ice changes specifically for individual low pressure systems (cyclones) observed during the melt period, we now want to consider the large-scale general atmospheric flow conditions.

**3.3.1 ZW3**

The monthly mean sea level pressure and meridional heat advection shown in Fig. 3 have a clear signal of ZW3, most strongly in August and October. Given these patterns, we examine the ZW3 index for the period of ice decay. Figure 7a shows the daily ZW3 index for 2013–2016. The years 2013–2015 are shown to put 2016 into perspective. Clearly, 2016 shows a distinctly different picture from

the preceding years 2013–2015. While in 2013–2015 the ZW3 index alternated between positive and negative values throughout the year, in 2016, the ZW3 index was almost continuously positive over a longer time period. The largest differences occur in the cold season, namely mid-April to June, and, with short interruptions, well into August. In 2016, the positive ZW3 index is predominant from mid-April far into July and, with short interruptions, until mid-October. The sudden and unexpected

decrease in SIE at the end of August comes after a prolonged period of positive ZW3 indices, whereas in November, when the record minimum SIE was observed, the ZW3 index is highly variable, both positive and negative. Given the influence of ZW3 on the sign and strength of the sensible heat flux between atmosphere and ocean and thus on sea ice growth or melt (Raphael, 2007), we suggest that the persistent presence of a positive ZW3 index in the winter months preconditioned the ice for melt.

Since an increased meridional exchange can also be related to ZW1, we looked at the 500-hPa geopotential height anomaly shown in Fig. 7b. It is created by subtracting the long-term zonal mean at each latitude, from the mean 500-hPa geopotential height field in November 2016, thus elucidating the asymmetry in the circulation suggested by the index. In August, as expected from the sign of the ZW3 index and the mean sea level pressure distribution in Fig. 6a, a distinct ZW3 pattern can be seen,

leading to fairly strong meridional flow and thus meridional heat exchange in the respective areas described in section 3.2.1. While in September the pattern is disturbed in the Atlantic sector (no

distinct meridional flow, see Fig. 4b), in October it reappeared, but was no longer as well-defined as in August. In November, while no clear ZW3 pattern is found, a distinct meridional flow can be seen around the Ross Sea and in the Weddell Sea in the first third of the month (see Fig. 4e). A clear ZW3
pattern is not expected here since the amplitude of the wave as shown in Fig. 7a is negative for most of the month. However, while a positive ZW3 index is not coincident with the decrease in November, its persistence in the preceding months will have preconditioned the ice so that the effect of any other activity that promotes ice loss would be amplified. The flow pattern associated with the strong negative anomaly in the 500-hPa geopotential height above the Weddell Sea is consistent with the
strong melt there due to warm air advection from the north.

Figure 8 illustrates the 2016 anomaly in the meridional heat flux for the entire year 2016 (Fig. 8a) as well as the period August to November 2016 only (Fig. 8b) with respect to the climatological mean of the period 1979–2015. Both the positive and the negative anomalies are stronger developed in the latter period, which agrees well with the prevailing ZW3 pattern described above.


### 3.3.2 Southern Annular Mode

Figure 9 shows the monthly mean SAM index based on the calculation of Marshall (2003) for 2016 and the climatological monthly means calculated for the period 1979–2015. In November, the months with the largest ice loss, the SAM index is highly negative, in fact, the second lowest SAM index since
the start of the satellite sea ice observations occurred. This low SAM index would suggest a rather meridional flow, resulting in a comparatively large meridional exchange of heat. Different from ZW3, which has been strongly positive already from May on and continuing during the entire winter months, thus hinting at increased meridional flow, the SAM index is positive in most months. However, as mentioned above, it should be kept in mind that SAM explains only part of the variability of the
circulation, and large regional differences in the flow pattern can occur. Additionally, the monthly means are not always meaningful since periods with positive and negative indices can cancel each other.

### 4    Discussion and conclusion

The record anomalies in Antarctic SIE in austral spring 2016 were associated with an unusually early start of the melt season with negative anomalies in SIA (SIE) observed as early as July (August). Although other components of the climate system, such as the ocean certainly contributed to this anomalous melt, our analysis suggests that increased meridional heat exchange with middle latitudes due to strongly meandering westerlies played a significant role. This anomalous circulation pattern is
expressed by a persistently positive ZW3 index, a measure of the meridionality of the flow, during

most of the winter and early spring, which was also discussed by Clem et al. (2017). The influence of ZW3 is reduced in November, whereas the phase of the SAM was strongly negative; its index reaching an almost record negative value. The first third of November was under the influence of strong warm air advection in the areas with negative sea ice anomalies. This relatively short, but intense, period triggered the accelerated ice melt, which was being continued in the following weeks due to the positive feed-back effects of the ice melt, i.e. lower albedo of the open vs. ice-covered ocean and increased turbulent heat flux from the ocean. The timing of this triggering was crucial for the increased melt; it happened exactly at the height of the melt season when conditions were set for melt anyway (strong radiation, high air temperatures).

Our results agree well with the general findings of Turner et al. (2017). While they stress the comparison of conditions in 2016 with the climatological means of amount and timing of SIE minima and maxima as well as mean location and intensity of cyclones, our study looks more closely at both the temporal and spatial evolution of SIA, SIE, and the development of the anomalies. In particular, we investigate the contribution of the different parts of the Southern Ocean to ice melt in more detail, and discuss the role of ice drift and the relationship between sea ice decay, SAM, and ZW3. Additionally, we found a clear relationship between the sea ice anomalies, low pressure systems, and meridional heat advection.

Three distinct anomalous melt periods were observed at the end of August, in the first half of September, in mid-October and in the first half of November, exhibiting SIC anomalies that are reminiscent of a ZW3 pattern. Four main areas of ice loss are found: The Amundsen-Bellingshausen Seas, the Weddell Sea, the Western Indian Ocean and the Western Pacific/Western Ross Sea. This loss is only partly counteracted by a positive anomaly in SIE off Marie Byrd Land, east of the Ross Sea (centered at approx. 140° W) and at the northeastern edge of the Weddell Sea. In November, the entire Weddell Sea strongly contributed to the accelerated sea ice retreat. The strong melting in the Amundsen-Bellingshausen Seas was caused by an extended and eastward-shifted ASL and consequent warm air advection at its eastern flank. The Western Pacific/Western Ross Sea experienced strong ice loss due to persistent warm air advection between September and November. The Western Indian Ocean exhibits negative anomalies in SIE and SIC, however, they do not increase significantly before November. The positive feedback mechanisms mentioned above further accelerated the sea ice decay. We argue that the quick development of most of these anomalies point towards an atmospheric origin of these anomalies. Two exceptions are probably the anomalies in the Western Indian Ocean and the Amundsen-Bellingshausen Seas that persisted throughout most of the winter and might have been preconditioned by warm anomalies in the previous year and reinforced by warm air advection during the austral fall 2016 (Stuecker et al., 2017).

The thermodynamic influence of atmospheric flow patterns described by positive ZW3 indices and negative SAM indices results in increased ice melt only shortly before and during the melt period, i.e.

when the air masses advected from the north have temperatures above the freezing point. A comparison of July SIE in the years 2009 and 2010, which had extremely different flow conditions (Schlosser et al., 2016), shows no significant difference in total SIE. While 2010 had extremely zonal flow conditions, 2009 exhibited strong amplification of Rossby waves with increased meridional heat exchange. The monthly mean SIE of July, however, was almost equal in those two years. In winter, atmospheric flow patterns are not sufficient to cause increased ice melt, since air temperatures are generally too low to cause ice melt, in spite of strong "warm" air advection.

Compaction of the ice by northerly atmospheric flow is only efficient if the SIC is clearly lower than 100 % or significant amounts of ridging and rafting occur. Northerly winds can in this case lead to a decrease of SIE along the ice edge and an increase in ice thickness towards the coast. The persistently positive ZW3 indices from mid-April to June (Fig. 7a) suggest that in 2016 the sea ice underwent the described preconditioning due to compaction. This process is also supported by the early decline in SIA in July.

In the northeastern Weddell Sea, positive SIC anomalies are found far into the melt season. These are probably caused by ice dynamics rather than thermodynamics, the strong westerlies in that area causing ice divergence and thus northeastward transport of the ice. Supporting thermodynamical influence is found in August and October, where a southerly flow with cold air advection from the continent prevailed at the western flank of a distinct low pressure system centered at the eastern border of the Weddell Sea. In contrast to the positive SIC anomalies around the ice edge in the northeastern Weddell Sea, the southeastern Weddell Sea experienced negative SIC anomalies, especially during August and November when the Weddell Sea polynya (also called Maud Rise polynya) reoccurred for the first time since the 1970s (Mazloff et al., 2017; Reid et al., 2017). This anomaly and most likely its reappearance in 2017 might be related to a strong regional sea ice divergence in 2016.

The results presented here are mostly qualitative. A regional atmospheric flow model combined with surface energy balance modelling would be necessary to quantify the ice melt in dependence on available energy due to warm air advection. The present study is focused on the atmospheric influence on the sea ice behavior in the melt season of 2016. The oceanic influence usually has longer time scales and reaction times, but cannot be neglected. Even though the memory effect is more important for the formation than for the melt of sea ice (Stammerjohn et al., 2012), there might be processes on a larger time scale that play a role here, too, especially with respect to the strong El Niño event in the previous year that might have preconditioned some of the anomalies due to the presence of warm surface waters in some regions (Stuecker et al., 2017). Our results suggest that the exceptional sea ice conditions in 2016 are owing to a superposition of several events in different parts of the Southern Ocean.

While the here discussed extreme decline of the Antarctic sea ice in austral spring 2016 is clearly unusual in the satellite record (since 1979), paleoclimatic records and modelling studies indicate that the natural variability of Antarctic sea ice might be substantially larger than the observed changes over the past decades (Hobbs et al., 2016; Jones et al., 2016). The large magnitude of the anomaly considerably affects the long-term trend of Antarctic SIE recent decades. Our analysis shows that including the period up to June 2017 leads to a reduction of the SIE trend by about one third, yielding $0.16\pm0.02$ million km$^2$ per decade compared to the 1979–2015 trend of $0.24\pm0.02$ million km$^2$ per decade. It remains to be seen how fast the Antarctic sea ice recovers from the unusual conditions in 2016 and whether those conditions will influence sea ice in the following years and even further reduce the long-term increase.

**Acknowledgements**

This study was financed by the Austrian Science Fund (FWF) under grant P28695. FAH was partly funded by the SNSF under grant number 175162. Sea ice data were provided by the NSIDC (National Snow and Ice Data Center), Boulder, CO (ftp://sidads.colorado.edu/DATASETS/NOAA/G02135/south/monthly/data/). ECMWF is greatly appreciated for the Interim reanalysis data (https://www.ecmwf.int/en/research/climate-reanalysis/era-interim). We are grateful to Gareth Marshall (BAS) for online provision of the observation-based SAM index (https://legacy.bas.ac.uk/met/gjma/sam.html).

**Author contribution**

The scientific analysis was jointly done by all authors, ES wrote the manuscript with contributions of MR and FAH; MR performed the ZW3 analysis, and FAH carried out all other calculations and graphics.

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

**Table 1.**

Regions used for the regional analysis (Fig. 5).

| EWS | Eastern Weddell Sea / Lazarev Sea | 20° W − 25° E |
|-----|-----------------------------------|----------------|
| WIO | Western Indian Ocean | 25° E − 85° E |
| EIO | Eastern Indian Ocean | 85° E − 130° E |
| WP | Western Pacific / Western Ross Sea | 130° E − 155° W |
| ERS | Eastern Ross Sea | 155° W − 125° W |
| ABS | Amundsen-Bellingshausen Seas | 125° W − 60° W |
| WWS | Western Weddell Sea | 60° W − 20° W |


**Figure captions**

**Figure 1.**

Monthly mean SIE anomalies since 1979 with respect to the 1979 to 2015 climatology. The period
1979 through 2015 is derived from the CDR NT data (green; Meier et al., 2013, updated 2016; provided by NSIDC). The period January 2015 through June 2017 is derived from the NRT data (red; Maslanik and Stroeve, 1999, updated daily; provided by NSIDC). Green shading shows the ±2 standard deviation of the CDR NT record and the green dashed line the respective long-term trend derived from linear regression analysis. The red dashed line shows the long-term trend for the period
1979 through June 2017 when extending the CDR NT record using the period January 2016 through June 2017 from the NRT record.

**Figure 2.**

Seasonal cycle of the daily **(a)** SIE, **(b)** SIA, and **(c)** sea ice area change, calculated as the temporal
derivative of SIA. The black lines show the climatological mean daily values for the period 1979 through 2015 derived from the CDR NT data (Meier et al., 2013, updated 2016; provided by NSIDC). The red lines show daily values for the year 2016 derived from the NRT data (red; Maslanik and Stroeve, 1999, updated daily; provided by NSIDC). (a and b) Grey shading shows the ±2 standard deviations of the CDR NT record and red shading the deficit of the year 2016 with respect to the CDR
NT record. (c) Positive (negative) SIA change is associated with a gain (loss) of SIA. Red and blue shading indicate periods in which the 2016 SIA deficit was growing and shrinking, respectively, with respect to the 1979 to 2015 period. The dashed line indicates the ±1 standard deviation of the CDR NT record.

**Figure 3.**

**(a–d)** Monthly mean SIC anomalies for the months August to November 2016 derived from the NRT data (red; Maslanik and Stroeve, 1999, updated daily; provided by NSIDC). Anomalies are calculated with respect to the climatological monthly mean derived from the CDR NT data from the period 1979 through 2015 (Meier et al., 2013, updated 2016; provided by NSIDC). Regions for the regional
analysis (Fig. 5) are indicated by the black lines and defined in Table 1. The gray and green contour lines denote the 2016 and the climatological (1979–2015) mean sea ice edge (15 % SIC), respectively.

**(e–h)** Monthly mean vertically integrated meridional heat fluxes (positive northward) from ERA-Interim (Dee et al., 2011) for the months August to November 2016.

**(i–l)** Monthly mean sea level pressure from ERA-Interim for the months August to November 2016.
Arrows indicate the approximate atmospheric flow direction.

**Figure 4.**

Mean vertically integrated meridional heat fluxes (positive northward) **(a)** and sea level pressure **(b)** from ERA-Interim (Dee et al., 2011) for the period 1–10 November 2016. The gray contour line denotes the mean sea ice edge (15 % SIC).

**Figure 5.**

As Fig. 2, but for each of the seven regions separately. Regions are indicated in the inset map and Fig. 3, and are defined in Table 1.

**Figure 6.**

Monthly mean sea ice drift (arrows) for the months August to November 2016 **(a to d)** derived from the low resolution sea ice drift product of OSI SAF (www.osi-saf.org; Lavergne, et al., 2010). White shading shows the respective monthly mean SIC derived from the NRT data (red; Maslanik and Stroeve, 1999, updated daily; provided by NSIDC). The grey contour line shows the 15 % SIC line.

**Figure 7.**

**(a)** Daily ZW3 index 2013–2016 (see section 2.3 for details).

**(b)** Monthly 500-hPa geopotential height zonal mean anomalies from ERA-Interim (Dee et al., 2011) for the period Aug–Nov 2016. Hatched areas indicate strongly negative values.

**Figure 8.**

Averaged monthly anomalies of the vertically integrated meridional heat fluxes (positive northward) from ERA-Interim for the year 2016 **(a)** and the period August to November 2016 **(b)** with respect to the climatological monthly means of the period 1979 to 2015. The gray contour line denotes the November 2016 monthly mean sea ice edge (15 % SIC).

**Figure 9.**

Monthly mean SAM index for 2016 and climatological monthly mean SAM index 1979–2015 (after Marshall, 2003).






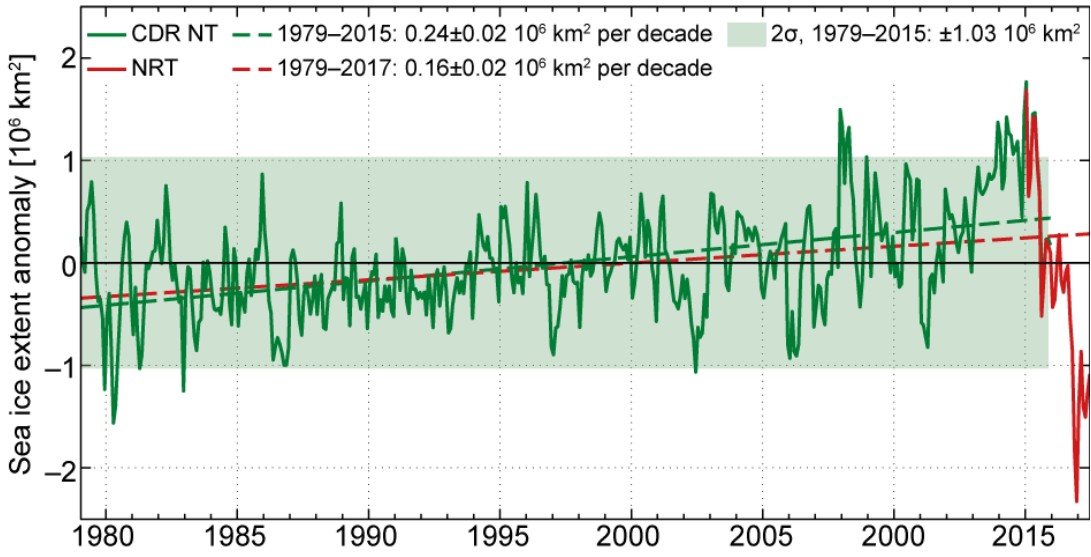

**Figure 1.**

Monthly mean SIE anomalies since 1979 with respect to the 1979 to 2015 climatology. The period
1979 through 2015 is derived from the CDR NT data (green; Meier et al., 2013, updated 2016; provided by NSIDC). The period January 2015 through June 2017 is derived from the NRT data (red; Maslanik and Stroeve, 1999, updated daily; provided by NSIDC). Green shading shows the ±2 standard deviation of the CDR NT record and the green dashed line the respective long-term trend derived from linear regression analysis. The red dashed line shows the long-term trend for the period 1979 through June 2017 when extending the CDR NT record using the period January 2016 through June 2017 from the NRT record.

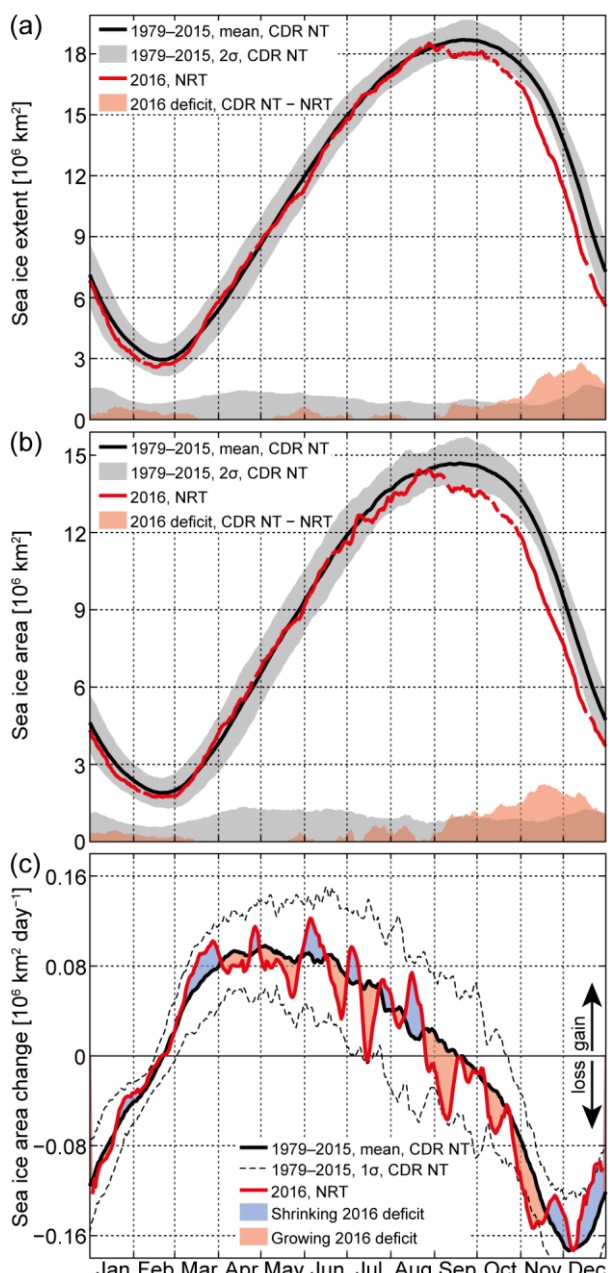

**Figure 2.**

Seasonal cycle of the daily **(a)** SIE, **(b)** SIA, and **(c)** sea ice area change, calculated as the temporal derivative of SIA. The black lines show the climatological mean daily values for the period 1979 through 2015 derived from the CDR NT data (Meier et al., 2013, updated 2016; provided by NSIDC). The red lines show daily values for the year 2016 derived from the NRT data (red; Maslanik and Stroeve, 1999, updated daily; provided by NSIDC). (a and b) Grey shading shows the ±2 standard deviations of the CDR NT record and red shading the deficit of the year 2016 with respect to the CDR NT record. (c) Positive (negative) SIA change is associated with a gain (loss) of SIA. Red and blue shading indicate periods in which the 2016 SIA deficit was growing and shrinking, respectively, with respect to the 1979 to 2015 period. The dashed line indicates the ±1 standard deviation of the CDR NT record.


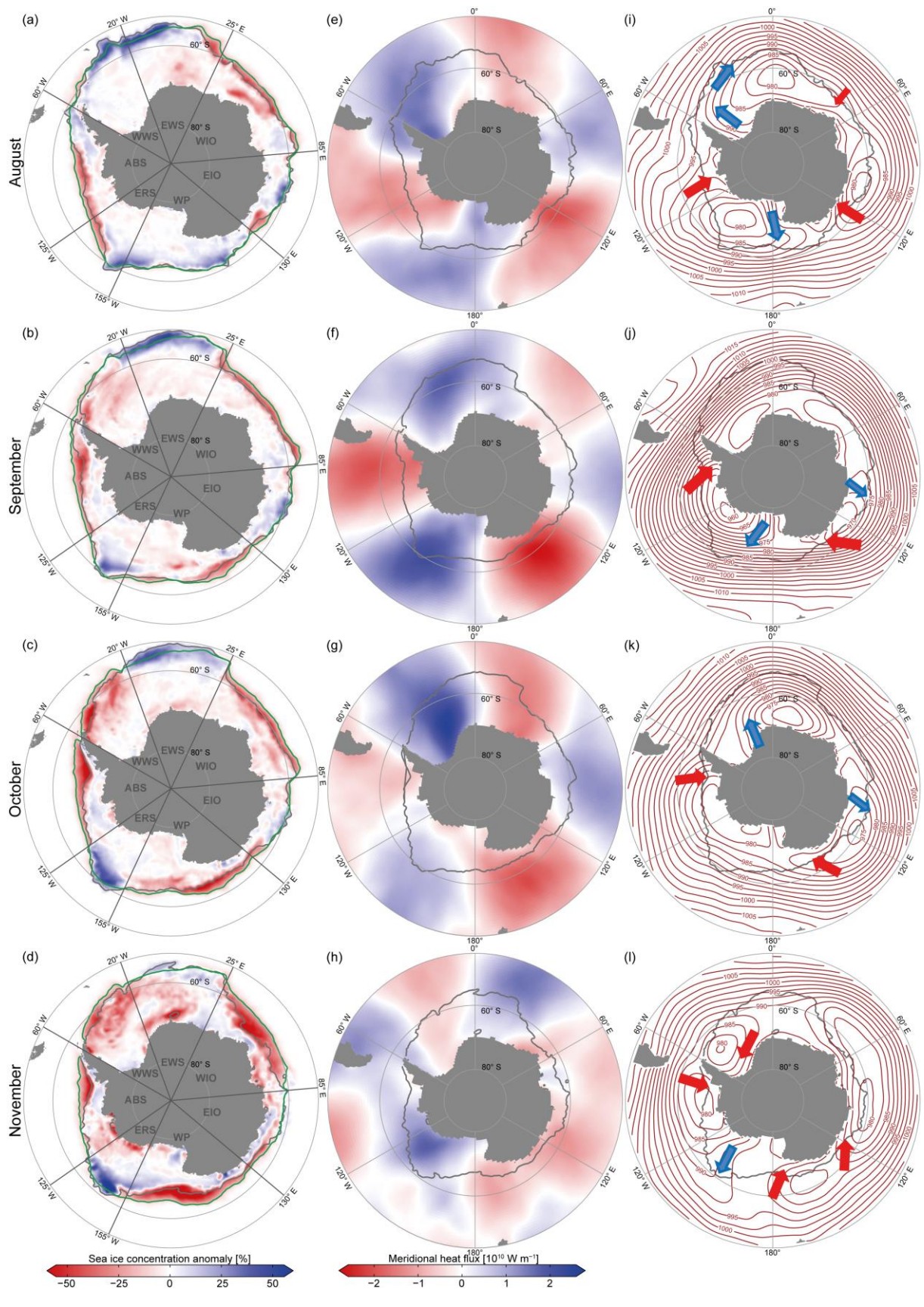

Sea ice concentration anomaly [%]

Meridional heat flux [10¹⁰ W m⁻¹]

**Figure 3.**

**(a–d)** Monthly mean SIC anomalies for the months August to November 2016 derived from the NRT
data (red; Maslanik and Stroeve, 1999, updated daily; provided by NSIDC). Anomalies are calculated
with respect to the climatological monthly mean derived from the CDR NT data from the period 1979
through 2015 (Meier et al., 2013, updated 2016; provided by NSIDC). Regions for the regional
analysis (Fig. 5) are indicated by the black lines and defined in Table 1. The gray and green contour
lines denote the 2016 and the climatological (1979–2015) mean sea ice edge (15 % SIC), respectively.

**(e–h)** Monthly mean vertically integrated meridional heat fluxes (positive northward) from ERA-
Interim (Dee et al., 2011) for the months August to November 2016.

**(i–l)** Monthly mean sea level pressure from ERA-Interim for the months August to November 2016.
Arrows indicate the approximate atmospheric flow direction.

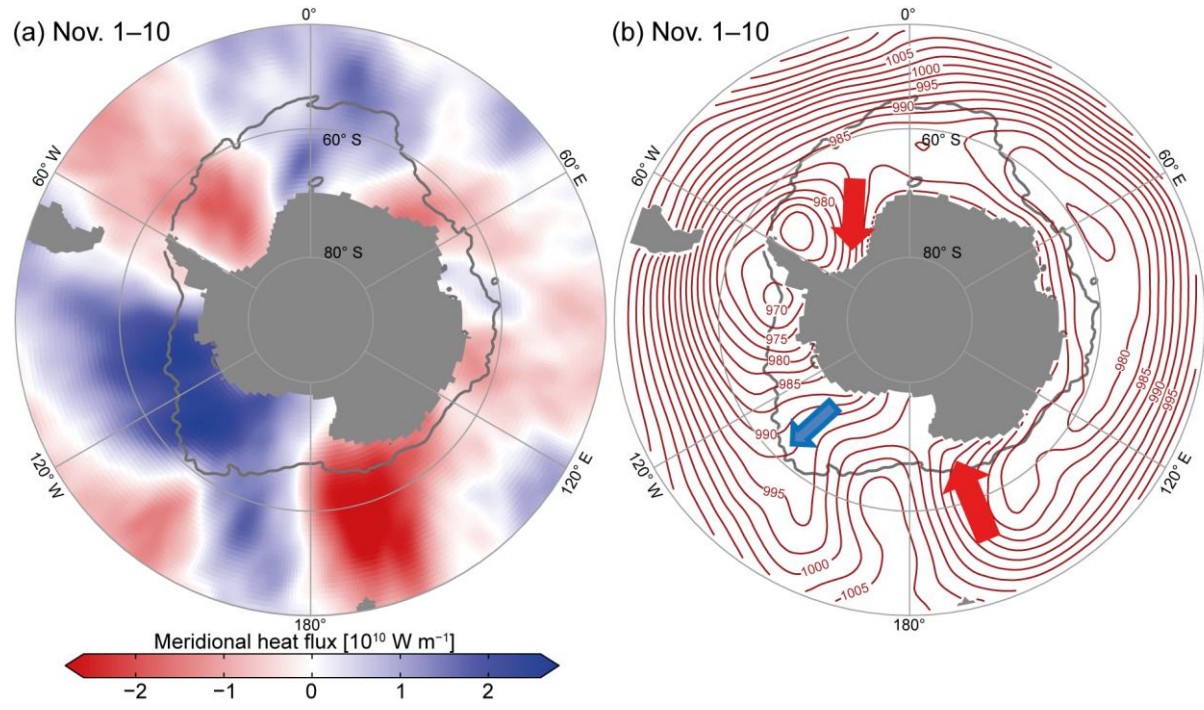


**Figure 4.**

Mean vertically integrated meridional heat fluxes (positive northward) **(a)** and sea level pressure **(b)** from ERA-Interim (Dee et al., 2011) for the period 1–10 November 2016. The gray contour line denotes the mean sea ice edge (15 % SIC).


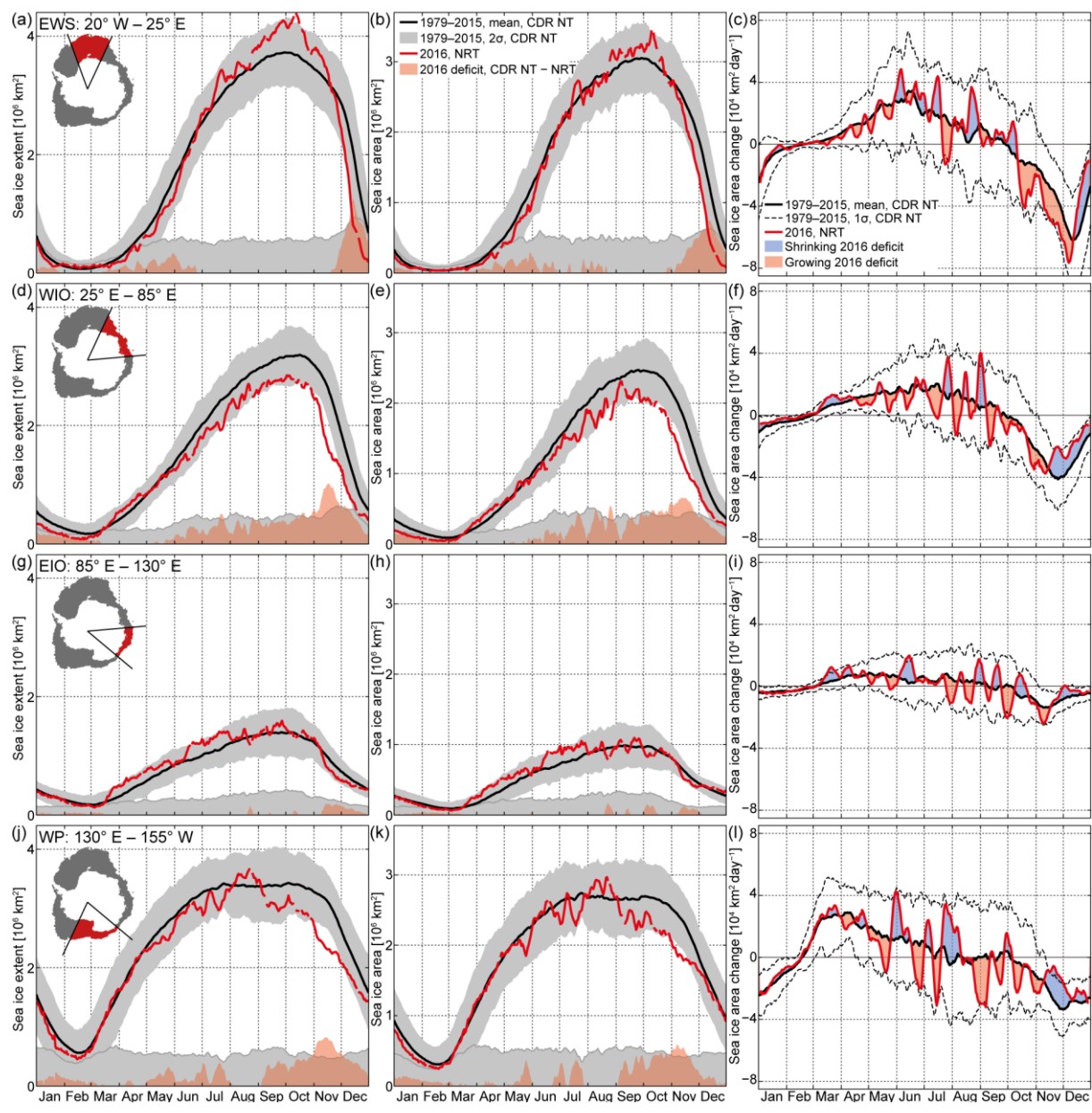

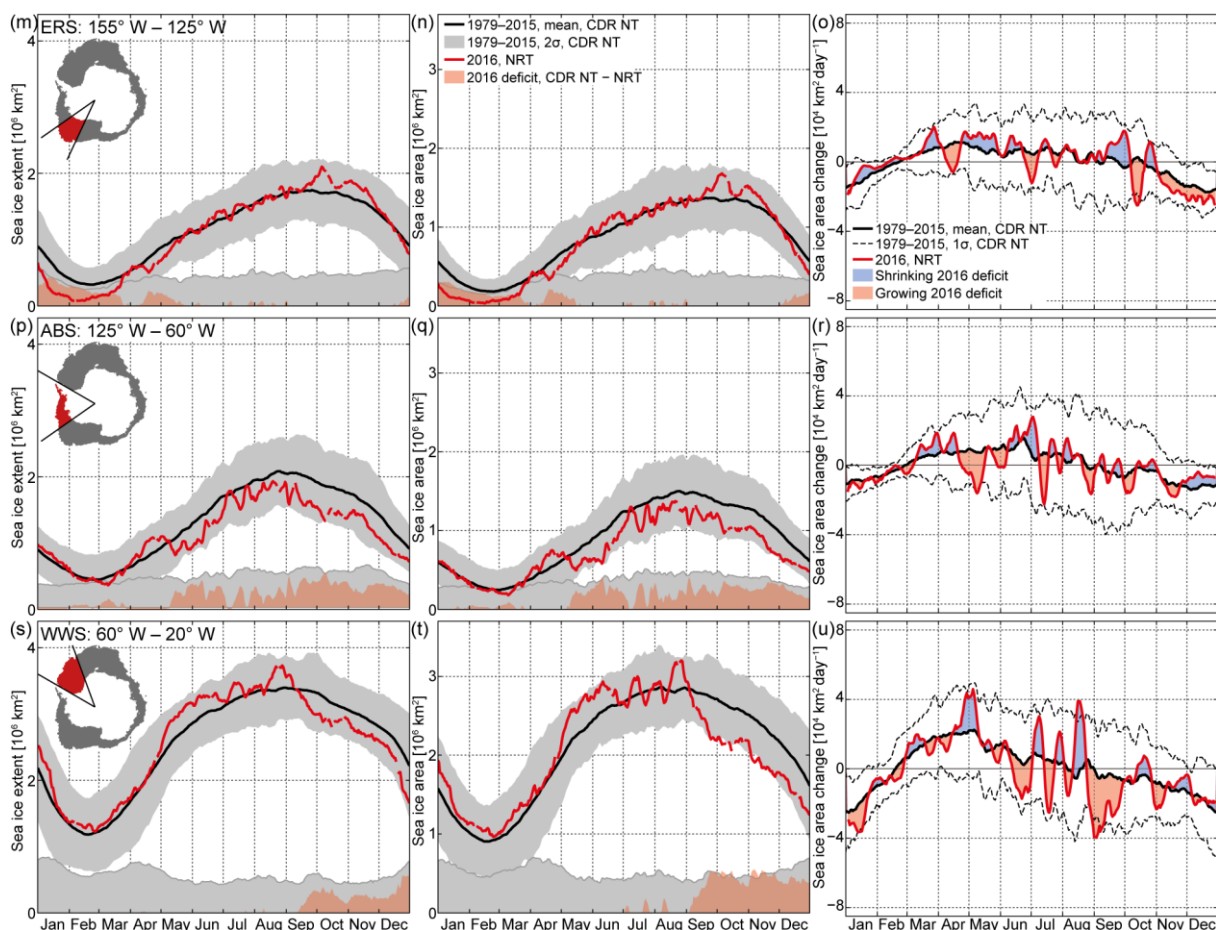

**Figure 5.**

As Fig. 2, but for each of the seven regions separately. Regions are indicated in the inset map and Fig.
3, and are defined in Table 1.

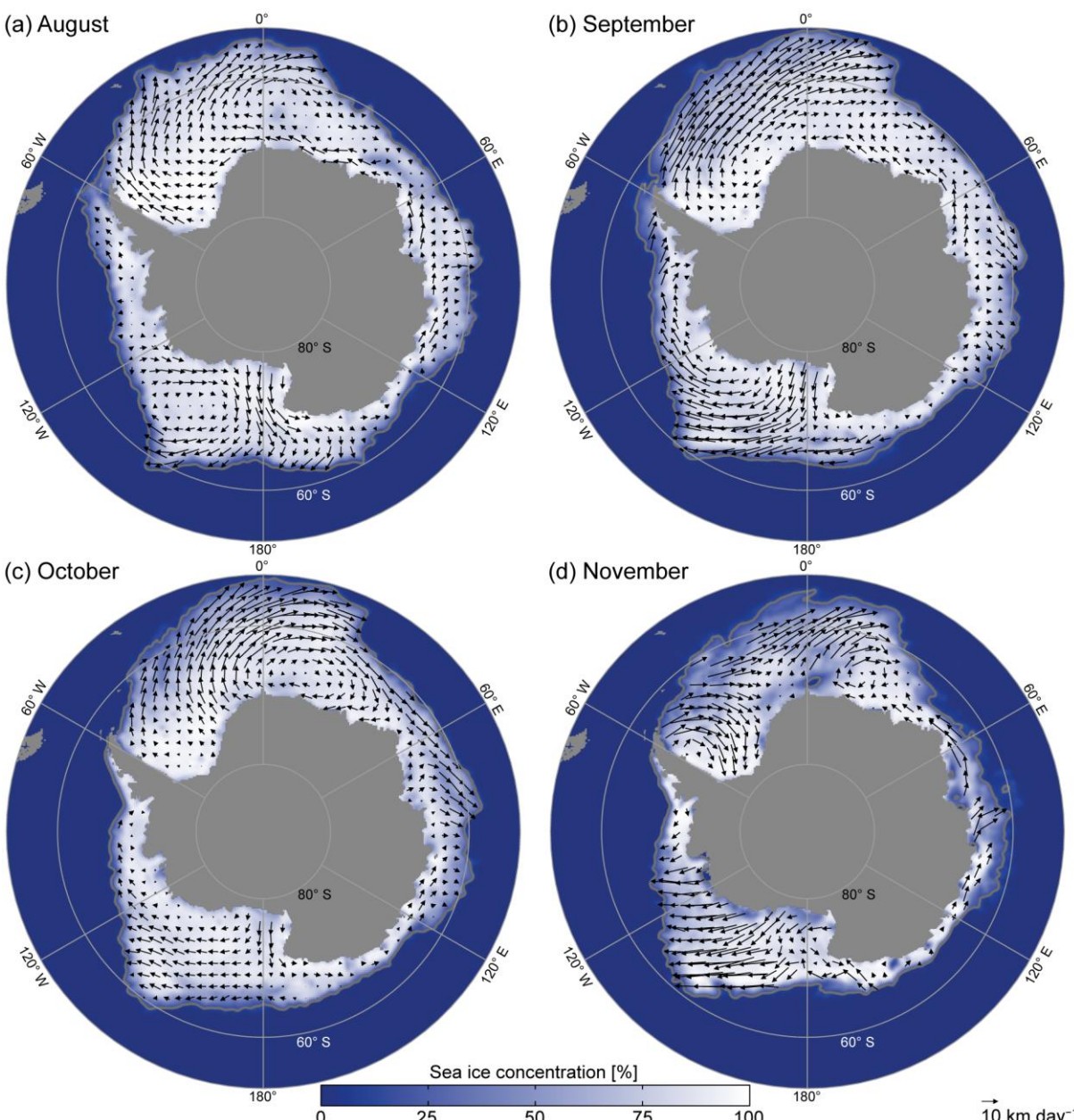

**Figure 6.**

Monthly mean sea ice drift (arrows) for the months August to November 2016 **(a to d)** derived from the low resolution sea ice drift product of OSI SAF (www.osi-saf.org; Lavergne, et al., 2010). White shading shows the respective monthly mean SIC derived from the NRT data (red; Maslanik and Stroeve, 1999, updated daily; provided by NSIDC). The grey contour line shows the 15 % SIC line.

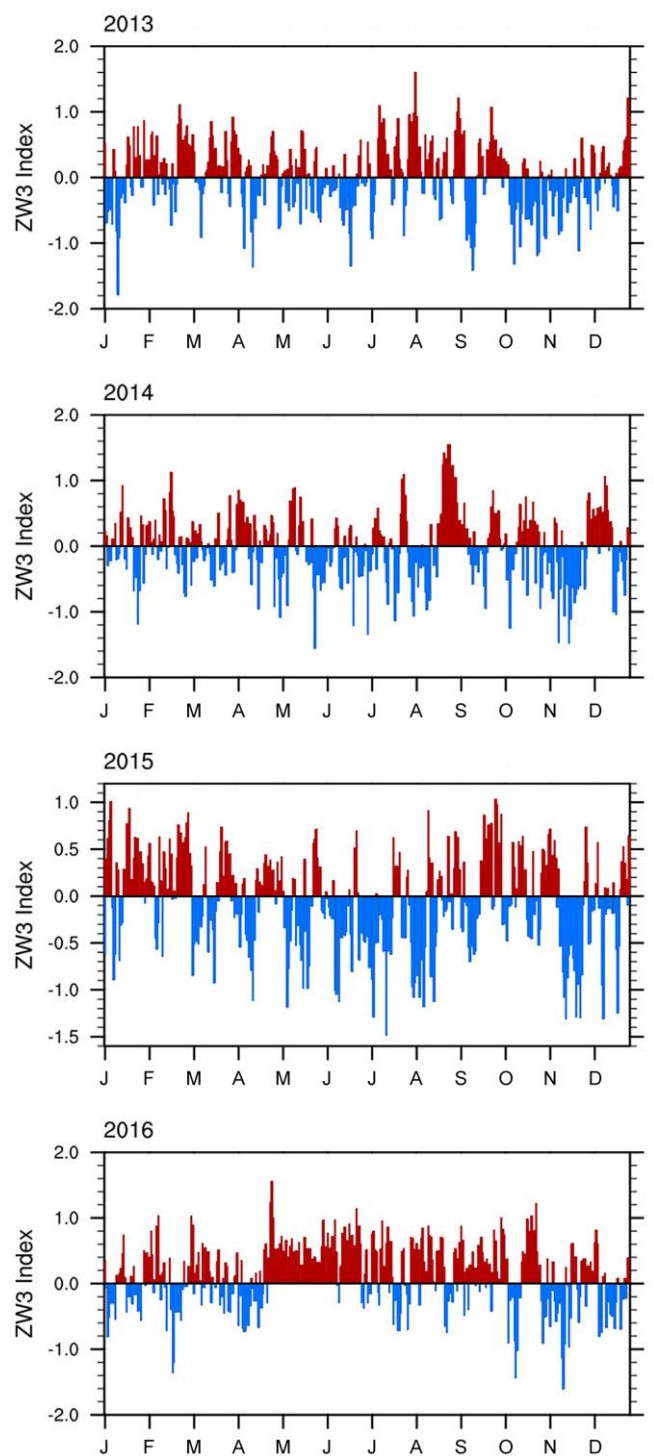

**Figure 7a.**

Daily ZW3 index 2013–2016 (see section 2.3 for details).

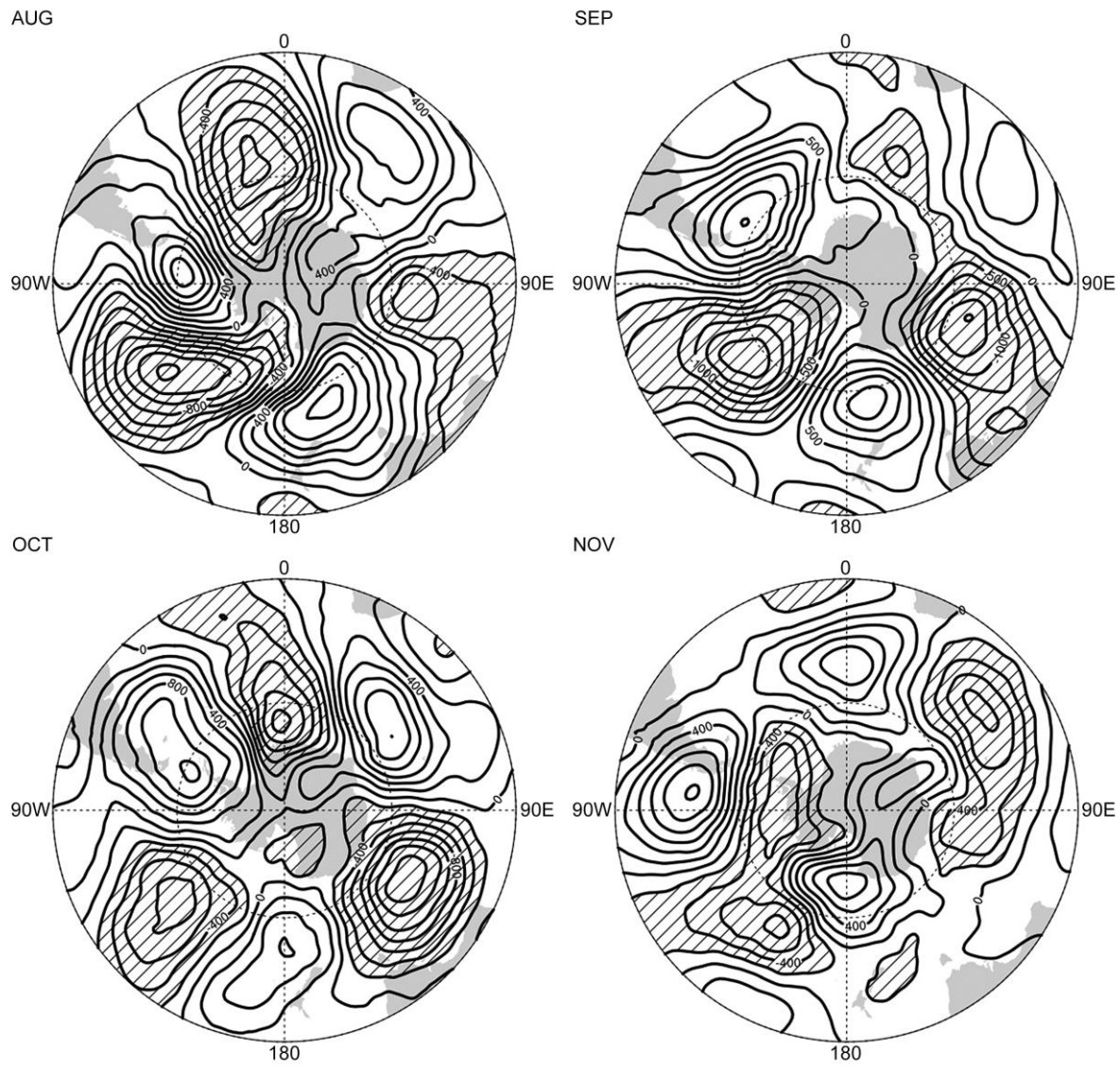

**Figure 7b.**

Monthly 500-hPa geopotential height zonal mean anomalies from ERA-Interim (Dee et al., 2011) for the period Aug–Nov 2016. Hatched areas indicate strongly negative values.

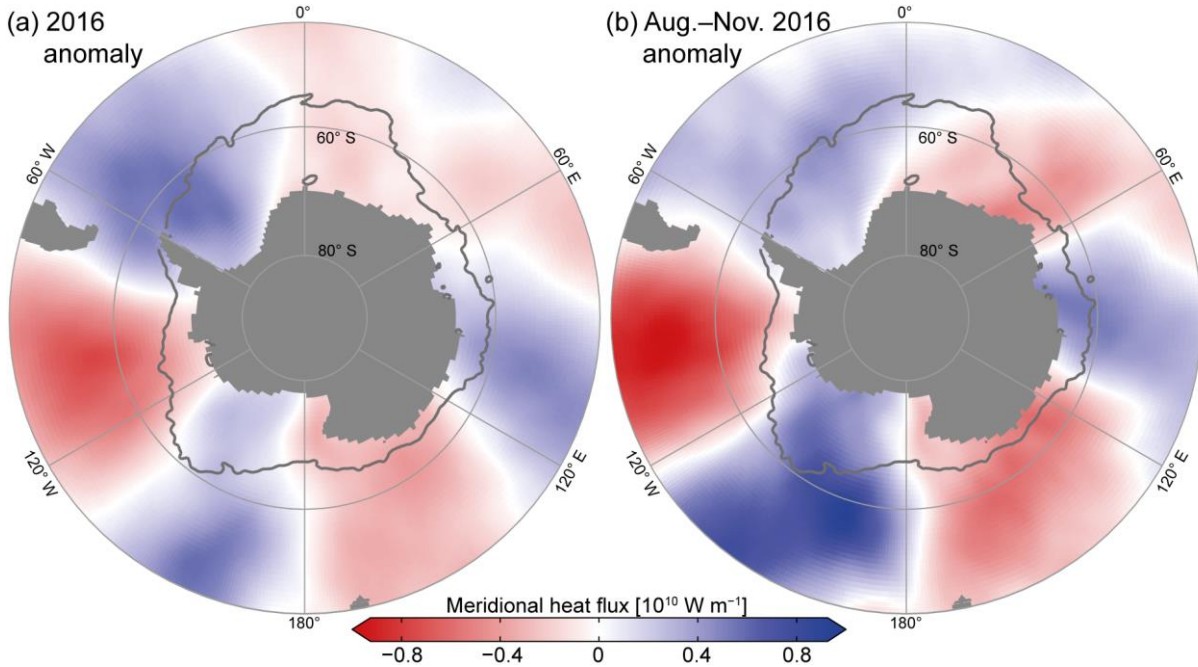

**Figure 8.**

Averaged monthly anomalies of the vertically integrated meridional heat fluxes (positive northward) from ERA-Interim for the year 2016 **(a)** and the period August to November 2016 **(b)** with respect to the climatological monthly means of the period 1979 to 2015. The gray contour line denotes the November 2016 monthly mean sea ice edge (15 % SIC).

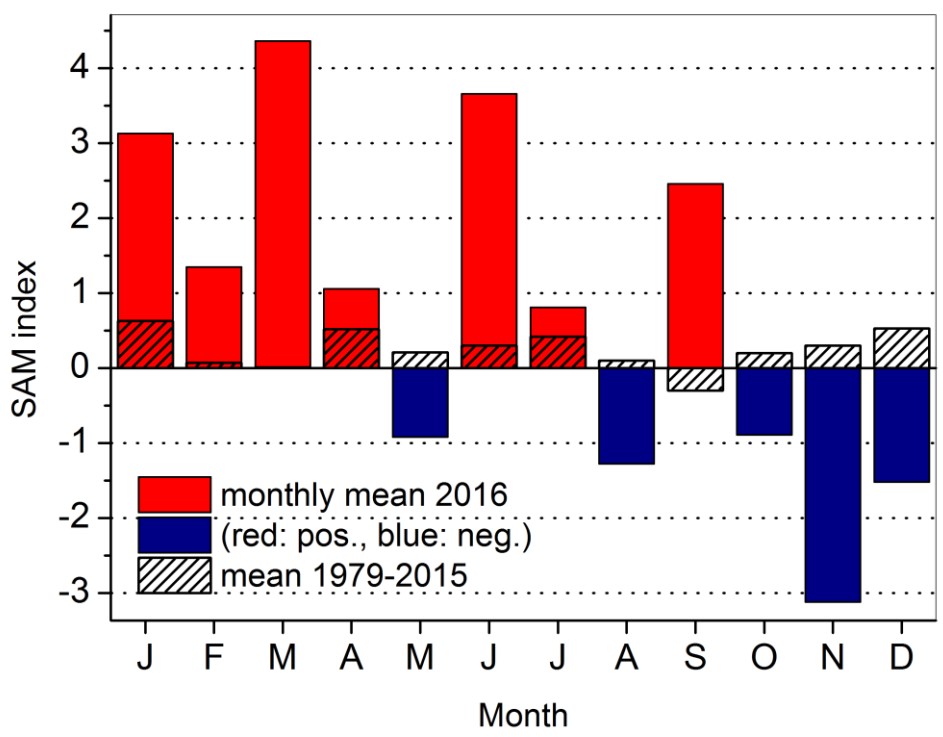

 **Figure 9.**

Monthly mean SAM index for 2016 and climatological monthly mean SAM index 1979–2015 (after Marshall, 2003).