# Peer review of "Atmospheric influences on the anomalous 2016 Antarctic sea ice decay"

_The Cryosphere, 2017_

## Referee Comment (RC1) · Anonymous Referee #1 · 6 Nov 2017

This paper discusses the atmospheric influences in the anomalously large variation of Antarctic sea ice observed in 2016. It claims that the early and large retreat of Antarctic sea ice was the result of atmospheric flow patterns, predominantly related to a zonal-wave three pattern until October 2016. Thereafter an atmospheric meridional flow during November, consistent with a negative SAM index, resulted in a "large meridional exchange of heat and moisture". It is generally well written with some nice analysis, although there are a number of missing references and citation errors.

The paper is quite descriptive. There is no direct objective analysis of atmospheric heat or moisture and no analysis discounting the influence of ocean surface temperatures playing their role in the 2016 event (as mentioned in line 127 regarding the paper by Stuecker). The analysis of zonal-wave three and SAM is similarly rather descriptive

and not conclusive. Without an objective analysis I would unfortunately suggest that the paper's claims are not substantiated. That said, I would encourage the authors to complete further analysis and resubmit the paper. More detailed discussion is included below.

Line 23 (and Line 59): "when combined with reduced Arctic SIE" – I don't think this is shown or referenced in the text at all, it just seems to be stated as a fact without proof.

Line 30: It might pay to be cautious in referring to the November 2016 SIE as "extraordinary". The relevance of the apparent sudden variability in net SIE (2014-2016) and the record low monthly SIC for November should perhaps be place into context with longer term variability (eg the past variability of SIE as shown in Hobbs et al 2016 and Jones et al 2016) in comparison to the relative short period of observations examined within this current paper. Would "...unprecedented low November SIE, based on the post 1979 satellite data." be more appropriate?

Line 51f: "reaching record extents in 2013, 2014 and 2015..." This should be "reaching record extents in 2012, 2013 and 2014...", and perhaps cite Reid and Massom, 2015 which covers all of 2012 through 2014.

Line 75: Please insert reference for Su 2017.

Line 81f: Please insert reference for Lee et al 2017.

Line 92: "Haumann , 2011" should be "Haumann, 2011".

Line 92: Note that Holland and Kwok's excellent study covers the relationship between wind and SIC trends for the period 1992-2010 and may not necessarily be extrapolated upon without caution. See Kwok et al 2017 for example – as referenced below.

Line 102: Is the term "Usually" based on data since 1979 and your Figure 1? If so then perhaps this should be stated, otherwise please cite some research showing this.

Line 124f: The sentence beginning with "Ross Sea and West..." probably needs rewrit-

ing.

Line 127ff: It probably should be noted that apart from Stuecker 2017, several BAMS State of the Climate sections mention 2016 Antarctic sea ice, please see the references below. In particular, Clem et al decomposes the atmospheric component, while Mazloff et al mention the ocean influence and Reid et al discuss the sea ice in general.

Line 182f: This sentence probably needs rewriting. Paragraph beginning at Line 243: Some of this paragraph confused me. There is mention of, for example, SIE in some regions being close to the long-term average – please see below for direct examples. I think perhaps you are confusing SIE with latitudinal extent, or otherwise could you please make this clearer.

Line 249f: "...with negative anomalies..." – negative anomalies of what?

Line 249f: "Largest negative anomalies". What is this a reference to: SIC, SIA, SIE or latitudinal extent? Note that SIE, as you have defined it, and latitudinal extent are two different things. If this is a reference to SIC or latitudinal extent (the contours on Figure 3?) then perhaps there should be reference to Figure 3a? If this is in reference to SIA or SIE then perhaps this should be shown in another way.

Line 251ff: The sentence beginning "In the Western..." should perhaps be rewritten. There is also some apparent confusion between what is showing "negative deviations" at approximately 130E and SIE – at least I'm confused!

Line 252: "approximately130" should be "approximately 130" – please insert a space.

Line 256: perhaps this should read, "However, in February 2017 the monthly mean SIA..." as otherwise this is a little ambiguous.

Line 267: "Hall and Visbeck should be 2002, not 2001?

Line 276ff: Other research (Clem et al 2017) has suggested that the SIC anomalies in this region are not necessarily related to atmospheric flow, and that they are related more to weaker ocean stratification and deeper convection over Maud Rise. The opening of the polynya in this region during August 2016 caused quite a bit of media attention.

Line 307: "This is shown in Figure 6e. . ." should probably be referring to Figure 4e?

Paragraph beginning at Line 314: There is a concluding summary here for section 3.2.1 that is not reflected directly by the analysis, and revolves around the words "strong warm air advection". Not once through the paper is atmospheric temperature directly analysed. There is some nice analysis and discussion within this paper, but much of the analysis that leads to this conclusion is descriptive rather than an objective analysis. Also, a number of papers (and you have cited one) have suggested that the 2016 Antarctic sea ice anomalies were possibly the result of a combination of atmosphere and ocean anomalies. Here you are suggesting the anomalies were the result of a warm atmosphere only, and I don't feel that you have directly shown this. You have not discounted the ocean impacts or objectively shown that a warm atmosphere specifically was responsible for the SIC anomalies. Given your concluding remarks and such a large anomaly in net SIE, surely you would be able to show that there was a large atmospheric temperature anomaly – or ocean surface anomaly?

Line 330: The sentence beginning, "The start of the melt period. . ." probably needs rewriting.

Line 335ff: From Figure 5u it would appear that R7 contributes significantly to the sea ice decay – in contrast to what is said here.

Line 348ff: From, "It shows generally good agreement. . .". Again, this is based on some quite subjective analysis.

Paragraph starting on Line 358: There is discussion here about the first third of the month of November being significant, and there is reference to Figure 4e. But there is no corresponding pattern of sea ice drift for this time period.

Line 361: "...leading to compaction..." and "...decreasing SIC" are not really consistent. Should you split this discussion into something like and inner and outer ice pack?

Line 374ff: ZW3 plots are given for the years 2013-2016, but with no real explanation as to why these years were chosen. ZW3 is put into perspective with the preceding three years, but there is no suggestion here that an August-October averaged ZW3 index of ~0.4 is significant or not. Is this ZW3 value 1, 2 or 3 standard deviations above normal for this time of the year, has it ever happened before and if so what was the consequence on the sea ice? Indeed, it would appear that for August 2014 the ZW3 was considerably well above that of 2016. In fact, looking at Raphael 2004 it would appear that there are long periods of positive ZW3, but there is no mention of the corresponding SIE, SIC for these years.

Line 444: Please reference Schlosser 2015, or should this be Schlosser 2016?

Line 510: There is no citation of Hobbs et al. Please remove or cite within text.

Line 517: There is no citation of Kottmeier and Sellmann 1996. Please remove or cite.

Line 545: Please remove or cite Peng et al.

Line 557: Please remove or cite Schlosser 1988.

Line 566ff: Please remove or cite both Simmonds papers.

Line 576ff: Note that Turner et al 2009 has been referenced twice – please remove one.

Line 581ff: Note that Turner et al 2014 should be Turner et al 2015.

General comments: Is the term Mio, used throughout the text in reference to "million", suitable for this Journal?

I suspect that within the text there are some instances where reference to Figure 3 and

[Figure]

Figure 4 get confused, or that perhaps that there should be reference to both Figures 3 and 4? For example, Lines 269, 281 and 291 discuss SIC anomalies but both times there are references to the figure showing MSLP (Figure 4).

"Figure" and "Fig" seem to be variously used through the text. Chose one of these that is appropriate for this Journal and be consistent through the text.

Table 1: There appear to be some discrepancies in the longitudes within this table.

Figure 3: What do the coloured contours represent on these figures?

Figure 4: I have some concerns with this figure. Are the arrows hand drawn and coloured – there is no mention in the text or in the figure caption as to how they are derived? They look rather subjective than objective.

Figure 7b: There is no mention of the hatching in this figure.

Suggested References:

Clem, K.R., S. Barreira, and R.L. Fogt, 2017: Atmospheric Circulation [in "State of the Climate in 2016"]. Bull. Amer. Meteor. Soc., 98 (8), S156–S158

Jones et al, 2016 Assessing recent trends in high-latitude Southern Hemisphere surface climate. Nature Climate Change 6, 917–926 (2016)

Kwok, R, et al 2017 Sea ice drift in the Southern Ocean: Regional patterns, variability, and trends. Elem Sci Anth, 5: 32, DOI: https://doi.org/10.1525/elementa.226

Mazloff, M.R., Sallée, J.B., Menezes V.V., Macdonald A.M., Meredith, M.,Newman, L., Pellichero V., Roquet F., Swart, S., Wahlin, A., 2017, State of the Southern Ocean in 2016, BAMS, 98 (8), S166-S167

Reid, P. & Massom, R. in State of the Climate in 2014 (ed. Blunden, J. & Arndt, D. S.) Spec. Suppl. Bull. .Am. Meteorol. Soc. 96, S163–S164 (2015).

Reid, P., S. Stammerjohn, R. A. Massom, J. L. Lieser, S. Barreira, and T. Scambos,

2017: Sea ice extent, concentration, and seasonality [in "State of the Climate in 2016"].
Bull. Amer. Meteor. Soc., 98 (8), S163–S166

---

## Author Comment (AC1) · 6 Nov 2017

We would like to express our gratitude to Ref. #1 for the thorough review. We will give a detailed response later. At this point, we just want to comment on two main points:

1. The reviewer states that we neglect oceanic influences. As the title suggests (and as we also mention in the text), our study focusses on the *atmospheric* influences on the sea ice behaviour. The *oceanic* influences are not a part of our study. However, in the discussion we mention that oceanic influences, namely on longer time scales, cannot be neglected.

2. The reviewer criticizes our study as "descriptive" and asks for an "objective" analysis.

Although our study is mainly qualitative (as stated in the discussion), it is clearly more than just "descriptive". It is a non-quantitative, but still objective analysis of the contribution of different areas to the total ice loss and investigates the prevailing synoptic situations associated with the sea ice retreat. There are clear physical relationships between the atmospheric flow patterns and the ice behaviour, which can explain to a large amount the atmospheric influence on the ice melt. A description would not contain any explanation and interrelations.

Naturally, we are aware of the fact that the atmosphere-ice-ocean system is highly complex and we agree that further, quantitative (modelling) studies would be necessary to fully understand the processes involved, both in the atmosphere and in the ocean. However, this was not the intention of our study.

---

## Short Comment (SC1) · 29 Nov 2017

**1 General Comments**

Schlosser et al. explain and investigate a range of atmospheric influences that lead to the anomalous sea ice decay in Antarctica in 2016. The rate of melt in this year was much higher compared with averages from previous years, at 2 Mio. km2, and there are many variables that can explain this. The paper establishes that some factors affecting the rate of ice decay include cyclonic activity, regional evolution of Sea Ice Extent (SIE) and Sea Ice Area (SIA), and atmospheric flow patterns. Mainly through analysis of the ZW3 (zonal wave 3) feedbacks and the SAM (Southern Annular Mode). This is significant as some of the research covered in this paper can be used to determine the exact

atmospheric influences on sea ice decay and how anthropogenic forcing can influence this, hence giving it relevance to climate change and other major controversies.There is a clear distinction between expected outcomes and unexpected outcomes in this paper. The anomalous results were clearly explained and identified, whilst the expected results were outlined from the beginning and both were used in the discussion of the atmospheric influences on sea ice decay.

**2 Specific Comments**

2.1 The authors clearly outline the aim of the paper, to discuss the possible reasons for the anomalous ice decay, in the introduction. The further explanation and previous work is written in clear and scientific language, however one suggestion would be that terms should be defined earlier. The terms ZW3 and SAM are used in the introduction with no explanation, and whilst they are covered later in the text this could be initially confusing.

2.2 In addition, some alternative ideas could be discussed, for example Stuecker et al. 2017 discuss the role of greenhouse gases and ozone forcing in addition to other atmospheric influences. Although great detail would not be relevant for this paper, mentioning these factors could help to link the results to other important parts of the cryosphere and give a wider picture of the papers significance.

2.3 Although relevant figures were used and they were well referenced in the text itself, having the figures and figure captions separate at the end of the paper was a little confusing. My suggestion would be to group them together and keep them at the end of the paper for ease of access.

2.4 Finally, the paper is well referenced and a large variety of literature was used. However most of these references were from before 2010 with one even dating back to 1902. One suggestion would be to use some more up to date references, as this is especially important considering the recent advances in this field of ice decay. The older referenes are not necessarily invalid however conditions and attitudes at the time

of these papers were much different from today, and this could be clarified within the paper.

2.5 In summary, the paper clearly analyses a wide range of available data and draws relevant hypotheses based on this, the sections are well laid out and clearly labelled and there is strong emphasis on past work and a wide range of references are cited. However, there are some minor errors that could be amended, therefore I suggest a review of the paper to correct these, after which it could be accepted.

3 Minor Corrections

3.1 Line 85- there is an unnecessary bracket (Enomoto and Ohmura (1990) should be (Enomoto and Ohmura, 1990)

3.2 Line 128- misspelled December as Dezember

3.3 Line 129- an accent is required for El Niño

3.4 Line 243- the comma after "to investigate" is unnecessary

3.5 Line 360-365- sentence is too long, needs to be broken up

3.6 Line 421- repeated word "accelerated ice melt melt"

3.7 Line 430- in my opinion this line would work better as a list of three i.e. "sea ice decay, SAM and ZW3" instead of repeating the ands

3.8 Line 445- again a matter of opinion but this could perhaps be broken into two sentences "... extremely zonal flow conditions. Whereas 2009 exhibited..."

References Stuecker M.F., Bitz C.M. and Armour K.C., 2017. Conditions leading to the unprecedented low Antarctic sea ice extent during the 2016 Austral spring season, Geophysical Research Letters, 44, 17, 9008-9019

---

## Author Comment (AC2) · 29 Nov 2017

**Response to Ref. #2**

We would like to express our gratitude to Sarah Cross for her careful and thorough review.

1 General Comments

Schlosser et al. explain and investigate a range of atmospheric influences that lead to the anomalous sea ice decay in Antarctica in 2016. The rate of melt in this year was much higher compared with averages from previous years, at 2 Mio. km2, and there are many variables that can explain this. The paper establishes that some factors affecting the rate of ice decay include cyclonic activity, regional evolution of Sea Ice Extent (SIE) and Sea Ice Area (SIA), and atmospheric flow patterns. Mainly through analysis of the ZW3 (zonal wave 3) feedbacks and the SAM (Southern Annular Mode). This is significant as some of the research covered in this paper can be used to determine the exact atmospheric influences on sea ice decay and how anthropogenic forcing can influence this, hence giving it relevance to climate change and other major controversies. There is a clear distinction between expected outcomes and unexpected outcomes in this paper.

The anomalous results were clearly explained and identified, whilst the expected results were outlined from the beginning and both were used in the discussion of the atmospheric influences on sea ice decay.

Specific Comments

2.1 The authors clearly outline the aim of the paper, to discuss the possible reasons for the anomalous ice decay, in the introduction. The further explanation and previous work is written in clear and scientific language, however one suggestion would be that terms should be defined earlier. The terms ZW3 and SAM are used in the introduction with no explanation, and whilst they are covered later in the text this could be initially confusing.

We agree that this might be confusing for readers not familiar with the subjects. We now give a short explanation where the terms occur for the first time in the "introduction and previous work" section (combined in the revised version) and leave the detailed explanation for the respective sections.

2.2 In addition, some alternative ideas could be discussed, for example Stuecker et al. 2017 discuss the role of greenhouse gases and ozone forcing in addition to other atmospheric influences. Although great detail would not be relevant for this paper, mentioning these factors could help to link the results to other important parts of the cryosphere and give a wider picture of the papers significance.

Stuecker et al. (2017) was published after we submitted our paper, we included it in the revised version. We also added some other points to the discussion, e.g. the occurrence of the Weddell Polynya (close to Maud Rise) in 2016 and teleconnections with ENSO.

2.3 Although relevant figures were used and they were well referenced in the text itself, having the figures and figure captions separate at the end of the paper was a little confusing. My suggestion would be to group them together and keep them at the end of the paper for ease of access.

We did this for the uploaded revised version. In the final version, the figures will be included in the text anyway in the usual TC layout.

2.4 Finally, the paper is well referenced and a large variety of literature was used. However most of these references were from before 2010 with one even dating back to 1902. One suggestion would be to use some more up to date references, as this is especially important considering the recent advances in this field of ice decay. The older references are not necessarily invalid however conditions and attitudes at the time of these papers were much different from today, and this could be clarified within the paper.

We agree. We added 15 new references in the revised version, of which 12 are not older than from 2012, 10 stem from the years 2016 and 2017.
Concerning the reference from 1902: We like to give credit to the researchers, who originally found the mentioned results, which was F. Nansen in this case. (Something that (in our (ES') opinion) tends to be neglected frequently nowadays, particularly when the reference is not available online.)

2.5 In summary, the paper clearly analyses a wide range of available data and draws relevant hypotheses based on this, the sections are well laid out and clearly labelled and there is strong emphasis on past work and a wide range of references are cited. However, there are some minor errors that could be amended, therefore I suggest a review of the paper to correct these, after which it could be accepted.

3 Minor Corrections

3.1 Line 85- there is an unnecessary bracket (Enomoto and Ohmura (1990) should be (Enomoto and Ohmura, 1990)

Corrected.

3.2 Line 128- misspelled December as Dezember

Corrected.

3.3 Line 129- an accent is required for El Niño

Corrected.

3.4 Line 243- the comma after "to investigate" is unnecessary

Corrected.

3.5 Line 360-365- sentence is too long, needs to be broken up

We re-wrote the entire paragraph, thus the sentence does not occur in the original form and length anymore (we agree that it had been too long and a bit hard to read).

3.6 Line 421- repeated word "accelerated ice melt melt"

Corrected.

3.7 Line 430- in my opinion this line would work better as a list of three i.e. "sea ice decay, SAM and ZW3" instead of repeating the ands

What we meant was the relationship between sea ice decay and SAM, and the relationship between sea ice decay and ZW3. However, of course, SAM and ZW3 are related, too, (if not as straightforward as it may seem, though), so we follow this advice.

3.8 Line 445- again a matter of opinion but this could perhaps be broken into two sentences ". . . extremely zonal flow conditions. Whereas 2009 exhibited. . ."
We broke this sentence into two sentences, as suggested.

References

Stuecker M.F., Bitz C.M. and Armour K.C., 2017. Conditions leading to the unprecedented low Antarctic sea ice extent during the 2016 Austral spring season, Geophysical Research Letters, 44, 17, 9008-9019

We included this reference.

---

## Author Comment (AC3) · 1 Dec 2017

**Response to Ref. #1:**

We would like to express our gratitude to Ref. #1 for the thorough and constructive review. His/her suggestions helped us to identify and resolve a number of weak or unclear points and formulations as well as to provide some additional analysis to support our conclusions. We will give a detailed response below.

Main points:

*1. This paper discusses the atmospheric influences in the anomalously large variation of Antarctic sea ice observed in 2016. It claims that the early and large retreat of Antarctic sea ice was the result of atmospheric flow patterns, predominantly related to a zonal-wave three pattern until October 2016. Thereafter an atmospheric meridional flow during November, consistent with a negative SAM index, resulted in a "large meridional exchange of heat and moisture". It is generally well written with some nice analysis, although there are a number of missing references and citation errors.*

We have included the suggested references and some additional ones to better reflect upon the current state of research in literature. We also carefully checked all citations and resolved all existing errors.

*2. The paper is quite descriptive. There is no direct objective analysis of atmospheric heat or moisture and no analysis discounting the influence of ocean surface temperatures playing their role in the 2016 event (as mentioned in line 127 regarding the paper by Stuecker). The analysis of zonal-wave three and SAM is similarly rather descriptive and not conclusive. Without an objective analysis I would unfortunately suggest that the paper's claims are not substantiated. That said, I would encourage the authors to complete further analysis and resubmit the paper. More detailed discussion is included below.*

As the title and text define, our study restricts itself and focusses on the *atmospheric* influences on the sea ice behaviour. The *oceanic* influences are not a part of our study, even though they are certainly an important factor. We re-wrote the introduction and the discussion to make it clearer that the topic of our study were the local atmospheric influences on the sea ice, but that the oceanic influence and teleconnections, which we have not been analysing here, play an important role, too.

Although our study is mainly qualitative (as stated in the discussion), it is not just "descriptive". It is a non-quantitative and objective analysis of the contribution of different areas to the total ice loss and investigates the prevailing synoptic situations and processes associated with the sea ice retreat. There are clear physical relationships between the atmospheric flow patterns and the ice behaviour, which can explain the atmospheric influence on the ice melt and its initiation. Our study contains an explanation and interpretation of the interrelations and identifies processes responsible for the 2016 anomalies and is therefore more than purely descriptive. Our analysis of the meridional heat transport is also "objective",

since there is a clear physical relation between the atmospheric circulation and meridional heat transport. In order to better illustrate this relation to the reader and provide a more quantitative measure of the meridional heat transport, we added a figure with the calculated vertically integrated meridional heat advection (from ERA-Interim) and combined it with Fig. 3 and Fig. 4. Then we discuss the SIC anomalies together with the advection field and the surface pressure field. We agree with the reviewer that we did not analyse the meridional moisture transport and therefore removed this from the related sentence in the results section

Naturally, we are aware of the fact that the atmosphere-ice-ocean system is highly complex and we agree that further, quantitative (modelling) studies would be necessary to better understand the processes involved, both in the atmosphere and in the ocean, and also their relative contributions. We extended this thought in the discussion. Such an investigation is, however, beyond the scope of our study.

*Line 23 (and Line 59): "when combined with reduced Arctic SIE" – I don't think this is shown or referenced in the text at all, it just seems to be stated as a fact without proof.*
We added a reference here in the introduction section. (it is highly unusual to give references in the abstract).

*Line 30: It might pay to be cautious in referring to the November 2016 SIE as "extraordinary". The relevance of the apparent sudden variability in net SIE (2014-2016) and the record low monthly SIC for November should perhaps be place into context with longer term variability (e.g. the past variability of SIE as shown in Hobbs et al 2016 and Jones et al 2016) in comparison to the relative short period of observations examined within this current paper. Would "...unprecedented low November SIE, based on the post 1979 satellite data." be more appropriate?*
We agree that this should be formulated more cautiously. Since we exceed the desired maximum number of 250 words in the abstract already, we changed this in the text accordingly and also added some remarks about the time period that was used for comparison in the discussion.

*Line 51f: "reaching record extents in 2013, 2014 and 2015 ..." This should be "reaching record extents in 2012, 2013 and 2014...", and perhaps cite Reid and Massom, 2015 which covers all of 2012 through 2014.*

We agree and corrected this and also included the suggested reference.

*Line 75: Please insert reference for Su 2017.*
Done.

*Line 81f: Please insert reference for Lee et al 2017.*
Done.

*Line 92: "Haumann , 2011" should be "Haumann, 2011".*

Done.

*Line 92: Note that Holland and Kwok's excellent study covers the relationship between wind and SIC trends for the period 1992-2010 and may not necessarily be extrapolated upon without caution. See Kwok et al 2017 for example – as referenced below.*
We included that Holland and Kwok (2012) only cover a sub-period of the satellite record. However, both the study by Haumann et al. (2014) and Kwok et al. (2017) show that meridional winds are also an important driver of long-term trends (entire satellite era; see their respective conclusions). Kwok et al. (2017) show that there are also regions were the relation between meridional winds and drift does not always hold, especially in coastal regions, but also argue that there might be issues with atmospheric reanalysis data in these coastal regions.

*Line 102: Is the term "Usually" based on data since 1979 and your Figure 1? If so then perhaps this should be stated, otherwise please cite some research showing this.*
Yes, we changed the formulation accordingly.

*Line 124f: The sentence beginning with "Ross Sea and West ..." probably needs rewriting.*
Done.

*Line 127ff: It probably should be noted that apart from Stuecker 2017, several BAMS State of the Climate sections mention 2016 Antarctic sea ice, please see the references below. In particular, Clem et al decomposes the atmospheric component, while Mazloff et al mention the ocean influence and Reid et al discuss the sea ice in general.*
We added all the BAMS citations and referred to them in the text.

*Line 182f: This sentence probably needs rewriting.*
Done.

*Paragraph beginning at Line 243: Some of this paragraph confused me. There is mention of, for example, SIE in some regions being close to the long-term average – please see below for direct examples. I think perhaps you are confusing SIE with latitudinal extent, or otherwise could you please make this clearer.*
We re-wrote this paragraph.

*Line 249f: "...with negative anomalies..." – negative anomalies of what?*
Of SIC, we changed this in the text.

*Line 249f: "Largest negative anomalies". What is this a reference to: SIC, SIA, SIE or latitudinal extent? Note that SIE, as you have defined it, and latitudinal extent are two different things. If this is a reference to SIC or latitudinal extent (the contours on Figure 3?) then perhaps there should be reference to Figure 3a? If this is in reference to SIA or SIE then perhaps this should be shown in another way.*
We reformulated this.

*Line 251ff: The sentence beginning "In the Western..." should perhaps be rewritten.*
Done.

*There is also some apparent confusion between what is showing "negative deviations" at approximately 130E and SIE – at least I'm confused!*
See above. We checked this and corrected this paragraph accordingly.

*Line 252: "approximately130" should be "approximately 130" – please insert a space.*
Done.

*Line 256: perhaps this should read, "However, in February 2017 the monthly mean SIA ..." as otherwise this is a little ambiguous.*
We don't understand the reviewer's point here, since the provided line number does not match the text. If the reviewer refers to line number 226 of the original manuscript, we agree that adding "February 2017" is less ambiguous and changed this sentence accordingly.

*Line 267: "Hall and Visbeck should be 2002, not 2001?*
We corrected this.

*Line 276ff: Other research (Clem et al 2017) has suggested that the SIC anomalies in this region are not necessarily related to atmospheric flow, and that they are related more to weaker ocean stratification and deeper convection over Maud Rise. The opening of the polynya in this region during August 2016 caused quite a bit of media attention.*
Thank you for pointing out that we did not really address the re-occurrence of the Weddell Sea polynya in 2016 and 2017. Actually, this is a very important point and we think that our study can at least partly explain its initial development in November 2016 due to a strong surface divergence, which might have triggered oceanic feedbacks now discussed in the manuscript. We included the polynya in the discussion and quoted the suggested reference.

*Line 307: "This is shown in Figure 6e" should probably be referring to Figure 4e?*
The Figures 3 and 4 have been combined now and the numbers changed accordingly.

*Paragraph beginning at Line 314: There is a concluding summary here for section 3.2.1 that is not reflected directly by the analysis, and revolves around the words "strong warm air advection".*
We removed this paragraph.

*Not once through the paper is atmospheric temperature directly analysed. There is some nice analysis and discussion within this paper, but much of the analysis that leads to this conclusion is descriptive rather than an objective analysis.*
We added a new figure with the northward heat advection from ECMWF-Interim Re-analysis to quantify our result discussed together with the surface pressure field and related winds. The meridional heat advection agrees very well with our qualitative results shown so far.

*Also, a number of papers (and you have cited one) have suggested that the 2016 Antarctic sea ice anomalies were possibly the result of a combination of atmosphere and ocean anomalies. Here you are suggesting the anomalies were the result of a warm atmosphere only, and I don't feel that you have directly shown this. You have not discounted the ocean impacts or objectively shown that a warm atmosphere specifically was responsible for the SIC anomalies. Given your concluding remarks and such a large anomaly in net SIE, surely you would be able to show that there was a large atmospheric temperature anomaly – or ocean surface anomaly?*

We did not state that the observed behaviour of the sea ice were the result of a warm atmosphere only and we agree that oceanic influences and feedback mechanisms might play an important role as well. Our study investigated the atmospheric influences only, but we also mentioned that the ocean is important, too. We reformulated the introduction and the discussion and added a number of studies addressing these issues to make this point clear, also including the effects of teleconnections with ENSO. Nevertheless, our analysis of the temporal evolution of the regional SIA and SIE anomalies imply an atmospheric origin of the anomalies in many regions that is associated with the warm air advection events and a rapid response of the SIA and SIE triggered by the event. We argue that if the ocean was initiating these anomalies, they would occur much more gradually and persist already over a longer time period. However, this gradual evolution of the anomalies occurs partly in the Western Indian Ocean sector and the Amundsen and Bellingshausen Seas, where it is likely that the ocean plays a critical role, as we now explicitly state.

*Line 330: The sentence beginning, "The start of the melt period..." probably needs rewriting.*
Done. .

*Line 335ff: From Figure 5u it would appear that R7 contributes significantly to the sea ice decay – in contrast to what is said here.*
We changed this accordingly.

*Line 348ff: From, "It shows generally good agreement...". Again, this is based on some quite subjective analysis.*
See above.

*Paragraph starting on Line 358: There is discussion here about the first third of the month of November being significant, and there is reference to Figure 4e. But there is no corresponding pattern of sea ice drift for this time period.*
We do not think it is necessary to show the ice drift for this period since melting is the predominant factor that influences sea ice decay during this period. However, we agree that the previous formulation was a bit confusing and might have implied that such a Figure existed. We rewrote this paragraph.

*Line 361: "...leading to compaction..." and "...decreasing SIC" are not really consistent. Should you split this discussion into something like and inner and outer ice pack?*
We agree that this was confusing and not quite correct. We corrected this.

*Line 374ff: ZW3 plots are given for the years 2013-2016, but with no real explanation as to why these years were chosen. ZW3 is put into perspective with the preceding three years, but there is no suggestion here that an August-October averaged ZW3 index of ~0.4 is significant or not. Is this ZW3 value 1, 2 or 3 standard deviations above normal for this time of the year, has it ever happened before and if so what was the consequence on the sea ice? Indeed, it would appear that for August 2014 the ZW3 was considerably well above that of 2016. In fact, looking at Raphael 2004 it would appear that there are long periods of positive ZW3, but there is no mention of the corresponding SIE, SIC for these years.*

The plots for 2013-2015 are shown to put 2016 into perspective. The index is normally so variable that it does not make much sense to calculate an average. 2016 is distinctly different from the preceding years. While in 2013-2015 the ZW3 index alternated between positive and negative values throughout the year, in 2016, the ZW3 index was almost continuously positive in the winter months. This is unusual and hints at a preconditioning of the sea ice for the later intense melt. We now added a new Figure 8, which shows the meridional heat flux anomalies of the year 2016 with respect to the period 1979-2015. The anomalies reveal a clear ZW3 pattern, which further supports our argument that the year 2016 was clearly exceptional during the period of the satellite record.

*Line 444: Please reference Schlosser 2015, or should this be Schlosser 2016?*
Yes, 2016, we corrected that.

*Line 510: There is no citation of Hobbs et al. Please remove or cite within text.*
We included this citation in the text.

*Line 517: There is no citation of Kottmeier and Sellmann 1996. Please remove or cite.*
We included this citation in the text.

*Line 545: Please remove or cite Peng et al.*
We removed it.

*Line 557: Please remove or cite Schlosser 1988.*
We removed it.

*Line 566ff: Please remove or cite both Simmonds papers.*
We cite both papers in the text now.

*Line 576ff: Note that Turner et al 2009 has been referenced twice – please remove one.*
Done.

*Line 581ff: Note that Turner et al 2014 should be Turner et al 2015.*
Corrected.

*General comments: Is the term Mio, used throughout the text in reference to "million",
suitable for this Journal?*

We replaced "Mio." by "million" accordingly to the journal's guidelines.

*I suspect that within the text there are some instances where reference to Figure 3 and
Figure 4 get confused, or that perhaps that there should be reference to both Figures
3 and 4? For example, Lines 269, 281 and 291 discuss SIC anomalies but both times
there are references to the figure showing MSLP (Figure 4).*

Since we combined Fig.3 and Fig. 4 in the revised version, we had to check all references to
those figures and corrected them accordingly.

*"Figure" and "Fig" seem to be variously used through the text. Chose one of these that
is appropriate for this Journal and be consistent through the text.*

Here we followed the journal's rule that „Figure" is not abbreviated when it occurs at the
beginning of a sentence, but it is abbreviated when it occurs in the middle of a sentence (see
https://www.the-cryosphere.net/for_authors/manuscript_preparation.html). We made sure that
all occurrences follow this rule.

*Table 1: There appear to be some discrepancies in the longitudes within this table.*

Thanks for pointing this out. We corrected that. (Some 1s had disappeared due to change of
column width.)

*Figure 3: What do the coloured contours represent on these figures?*

The green line represents the monthly mean SIE 1979-2015 the black line the SIE in 2016 for
the corresponding month. We added this information in the figure caption and in the text,
where it occurs for the first time.

*Figure 4: I have some concerns with this figure. Are the arrows hand drawn and
coloured – there is no mention in the text or in the figure caption as to how they are
derived? They look rather subjective than objective.*

Since our publication was submitted to The Cryosphere, we address a readership that not
always has knowledge about atmospheric dynamics and even meteorologists get a bit mixed
up sometimes when it comes to clockwise or anti-clockwise rotation on the Southern
Hemisphere, we tried to facilitate the reading of the figure by adding arrows that indicated the
warm or cold air advection. The arrows agree well with the blue and red areas in our new
figure of meridional heat advection, the latter not explaining the reason for the advection,
though. Thus, we would like to keep the figure with the pressure fields, including the
illustrating arrows.

We added an explanation of the arrows in the discussion of the new Fig. 3.

*Figure 7b: There is no mention of the hatching in this figure.*

We added this information.

*Suggested References:*

*Clem, K.R., S. Barreira, and R.L. Fogt: Atmospheric Circulation [in "State of the Climate in 2016"]. Bull. Amer. Meteor. Soc., 98 (8), S156–S158, 2017.*

*Jones et al, 2016 Assessing recent trends in high-latitude Southern Hemisphere surface climate. Nature Climate Change 6, 917–926 (2016), 2016.*

*Kwok, R, et al 2017 Sea ice drift in the Southern Ocean: Regional patterns, variability, and trends. Elem Sci Anth, 5: 32, DOI: https://doi.org/10.1525/elementa.226*

*Mazloff, M.R., Sallée, J.B., Menezes V.V., Macdonald A.M., Meredith, M., Newman, L., Pellichero V., Roquet F., Swart, S., Wahlin, A.. State of the Southern Ocean in 2016, BAMS, 98 (8), S166-S167, 2017.*

*Reid, P. & Massom, R. in State of the Climate in 2014 (ed. Blunden, J. & Arndt, D. S.) Spec. Suppl. Bull. .Am. Meteorol. Soc. 96, S163–S164 (2015).*

*Reid, P., S. Stammerjohn, R. A. Massom, J. L. Lieser, S. Barreira, and T. Scambos, Sea ice extent, concentration, and seasonality [in "State of the Climate in 2016"]. Bull. Amer. Meteor. Soc., 98 (8), S163–S166, 2017.*

Thank you for the additional references. We included all of them.

---

## Referee Comment (RC2) · Anonymous Referee #2 · 5 Dec 2017

Review on "Atmospheric influences on the anomalous 2016 Antarctic sea ice decay" by Elisabeth Schlosser, F. Alexander Haumann, and Marilyn N. Raphael.

General Comments The study is well written, well organized, and a joy to read. However, I do find the study to be a bit too much on the qualitative/descriptive side with many of the claims made on how the atmosphere "influenced" the sea ice being a bit speculative. Furthermore, I am not sure if we are learning anything new here. As stated by Referee 1, many of the descriptive details surrounding the sea ice, atmospheric circulation, and SAM pattern during 2016 are already discussed in the State of the Climate in 2016 Antarctica chapter. Without more quantification of mechanisms, such as quantifying advection, melt, and the role of the ocean, I don't see what new information is being presented here. I also strongly encourage the authors to place their

findings more in context with other work, particularly Turner et al. (2017). I recommend the authors perform a major revision and resubmit at a later time.

Specific Comments There is a lot of referencing to place names (particularly ocean basins and seas) throughout the study, and so I recommend the authors include a map to go along with Table 1. I also recommend giving new names to the regions (R1, R2, etc) so there is some connection to their respective geographic place names (e.g., western Ross Sea as wRS, etc). This will also reduce instances of referring to both the place name and the respective "region" name for clarification in the text (for example, lines 329-331), which is confusing and makes the R1, R2, etc. names seem unnecessary. If sensible region names are defined, they could be used throughout the manuscript without requiring further clarification.

Line 116: Please add citation Meehl et al. (2016) and their finding that tropical Pacific variability also influence meridional winds and associated sea ice extent.

Line 117-118: As you mention below in lines 125-127, Turner et al. (2017) already established northerly wind/warm air advection was a major contributor to the 2016 record sea ice loss. What are we learning here that we don't already know?

Line 245-246: How does adding two extra sub-areas compare/expand upon the results of Turner et al. (2017)? Please make these new insights clear by placing them into context of Turner et al. (2017).

Figure 3: Please specify in the caption what the green and grey lines are. I assume green is the average SIE and grey is the 2016 SIE, but it needs to be specified.

Line 255-256: The negative SIC anomalies in the Amundsen and Bellingshausen Seas actually appear quite similar in magnitude to those in the Indian Ocean. Without quantifying this, I don't think it can be said here.

Line 309-310 and 314-316: Although I appreciate the schematic arrows, without quantifying advection there is no way of determining that warm air advection explained any

portion of the sea ice loss. Furthermore, actual surface air temperature over the sea ice would likely need to be analyzed to determine if, even in the presence of warm air advection, temperatures were actually warm enough to melt the ice as the authors claim.

Lines 332-334: This seems highly speculative.

Please add DOIs to bibliography

Technical Corrections Line 94: SIC has not been defined. Please define it here and use SIC for the remainder of the study

Line 124: remove "were"

Line 128: change to "December"

Line 130: remove "rather"

Line 137: ECMWF is never defined

Line 163: change "today" to "present"

Line 205: please clarify what "Mio." Means

Line 210: no longer need to continue defining SIC, SIE, SIA as they are already defined

Line 259: Change "Figure 6" to "Figure 4", and please clarify whether this is sea level pressure (as stated in caption) or surface pressure (as stated in text)

Line 269-270: Already defined as the Amundsen Sea Low / ASL, so just use ASL here

Line 275: Please remove the words "masses" and "right"

Line 288: Would say ASL instead of "Pacific low"

Line 305: Just put "periods" in parenthesis

Line 307: Please change to Figure 4e

References Meehl, G. A., J. M. Arblaster, C. M. Bitz, C. T. Y. Chung, and H. Seng, 2016: Antarctic sea-ice expansion between 2000 and 2014 driven by tropical Pacific decadal climate variability. Nature Geoscience 9, 590–595, doi:10.1038/ngeo2751

---

## Author Comment (AC4) · 3 Jan 2018

**Response to Ref #2**

We would like to express our gratitude to Ref. #2 for the thorough and constructive review. His/her comments helped us to identify and resolve a number of weak or unclear points and formulations as well as to provide some additional analysis to support our conclusions. We will give a detailed response below.

*General Comments*

*The study is well written, well organized, and a joy to read. However, I do find the study to be a bit too much on the qualitative/descriptive side with many of the claims made on how the atmosphere "influenced" the sea ice being a bit speculative. Furthermore, I am not sure if we are learning anything new here. As stated by Referee #1, many of the descriptive details surrounding the sea ice, atmospheric circulation, and SAM pattern during 2016 are already discussed in the State of the Climate in 2016 Antarctica chapter. Without more quantification of mechanisms, such as quantifying advection, melt, and the role of the ocean, I don't see what new information is being presented here. I also strongly encourage the authors to place their findings more in context with other work, particularly Turner et al. (2017). I recommend the authors perform a major revision and resubmit at a later time.*

As we stated already in our response to Ref#1, although our study is mainly qualitative (as stated in the discussion), it is not just "descriptive". It is a non-quantitative and objective analysis of the contribution of different areas to the total ice loss and investigates the prevailing synoptic situations and processes associated with the sea ice retreat. There are clear physical relationships between the atmospheric flow patterns and the ice behaviour, which can explain the atmospheric influence on the ice melt and its initiation. Our study contains an explanation and interpretation of the interrelations between atmospheric circulation and sea ice anomalies and identifies processes responsible for the 2016 anomalies. Therefore, it is more than purely descriptive. Our analysis of the meridional heat transport is also not speculative, since there is a clear physical relation between the atmospheric circulation and meridional heat transport. In order to better illustrate this relation to the reader and provide a more quantitative measure of the meridional heat transport, we added a figure with the calculated vertically integrated meridional heat advection (from ERA-Interim) and combined it with Fig. 3 and Fig. 4. Then we discuss the SIC anomalies together with the advection field and the surface pressure field. Our previous qualitative results were strongly supported by the quantified meridional advection.

Compared to Turner et al. (2017) and the BAMS paragraphs, we not only quantify the advection now, we also analyse the temporal and spatial development of the sea ice decline in considerably more depth and detail than the previous studies and investigate ZW3 in more detail. In addition, we analyze the sea ice area change (temporal derivative) of the daily sea ice area, which provides new insights into the development of the anomalies.

Naturally, we are aware of the fact that the atmosphere-ice-ocean system is highly complex and we agree that further, quantitative (modelling) studies would be necessary to better understand the processes involved, both in the atmosphere and in the ocean, and their relative contributions.

We extended this thought in the discussion. Such an investigation is, however, beyond the scope of our study.

*Specific Comments:*
*There is a lot of referencing to place names (particularly ocean basins and seas) throughout the study, and so I recommend the authors include a map to go along with Table 1. I also recommend giving new names to the regions (R1, R2, etc.) so there is some connection to their respective geographic place names (e.g., western Ross Sea as wRS, etc.). This will also reduce instances of referring to both the place name and the respective "region" name for clarification in the text (for example, lines 329-331), which is confusing and makes the R1, R2, etc. names seem unnecessary. If sensible region names are defined, they could be used throughout the manuscript without requiring further clarification.*

Thank you for this suggestion. We agree that the suggested naming of the sub-regions would make it easier for the reader to follow. Accordingly, we changed them to EWS, WIO, EIO, WP, ERS, ABS, and WWS.

*Line 116: Please add citation Meehl et al. (2016) and their finding that tropical Pacific variability also influence meridional winds and associated sea ice extent.*
Done.

*Line 117-118: As you mention below in lines 125-127, Turner et al. (2017) already established northerly wind/warm air advection was a major contributor to the 2016 record sea ice loss. What are we learning here that we don't already know?*

The study by Turner et al, as the purpose of GRL publications is defined, provides quick information about a recent topic in a relatively brief publication. They describe in relative detail the climatological behaviour of the sea ice and then try to explain the features observed in 2016. Our study investigates the temporal and spatial development of SIA and SIE in considerably more detail and depth than a short GRL paper can do. We also added the meridional heat advection for quantification now. Turner et al. use more general phrases, whereas we give detailed information to all single sub-areas and time periods that show that the situation is much more complex than described earlier, i.e. where and why did the warm air advection occur and what was the response of the sea ice. For example, Turner et al. wrote "rapid ice retreat in the Weddell Sea took place in strong northerly flow after an early maximum ice extent in late August". We show that the western and eastern parts of the Weddell Sea behaved differently (thus more sub-regions than in the Turner paper), and particularly the Eastern Weddell Sea did not contribute substantially to the ice retreat before November. (the western Weddell Sea not before October). Equally the BAMS chapters are rather short and less detailed than our study.

*Line 245-246: How does adding two extra sub-areas compare/expand upon the results of Turner et al. (2017)? Please make these new insights clear by placing them into context of Turner et al. (2017).*

Our definition of sub-areas closely followed the observed sea ice anomalies and not a climatology established earlier. This very specific definition of sub-areas allowed a detailed investigation and explanation of the observed phenomena and the evolution of the sea ice anomaly. Especially the sub-division of the Weddell Sea was necessary due to the different behaviour of the eastern and western Weddell Sea. We added this reasoning in the text.

We discussed Turner et al. in more detail in the discussion now and explained the differences to our study: Our results agree well with the general findings of Turner et al. (2017), but give more details due to the higher spatial resolution used in our study and also quantify the meridional warm air advection that was discussed only qualitatively by Turner et al. (2017). While they stress the comparison of conditions in 2016 with the climatological means of amount and timing of SIE minima and maxima as well as mean location and intensity of cyclones, our study looks more closely at both the temporal and spatial evolution of SIA and SIE, investigates the contribution of the different parts of the Southern Ocean to ice melt in more detail, discusses the role of ice drift and the relationship between sea ice decay, SAM and ZW3.

*Figure 3: Please specify in the caption what the green and grey lines are. I assume green is the average SIE and grey is the 2016 SIE, but it needs to be specified.*

We explained this now in the caption and also in the text.

*Line 255-256: The negative SIC anomalies in the Amundsen and Bellingshausen Seas actually appear quite similar in magnitude to those in the Indian Ocean. Without quantifying this, I don't think it can be said here.*

We reformulated this and referred to the position of the sea ice edge rather than SIC. The differences can be clearly seen in Fig. 3.

*Line 309-310 and 314-316: Although I appreciate the schematic arrows, without quantifying advection there is no way of determining that warm air advection explained any portion of the sea ice loss. Furthermore, actual surface air temperature over the sea ice would likely need to be analysed to determine if, even in the presence of warm air advection, temperatures were actually warm enough to melt the ice as the authors claim.*

See above: following also the advice of Ref.#1, in order to provide a more quantitative measure of the meridional heat transport, we added a figure with the calculated vertically integrated meridional heat advection (from ERA-Interim) and combined it with Fig. 3 and Fig. 4. Then we discuss the SIC anomalies together with the advection field and the surface pressure field. Our arrows agree very well with the areas of warm and cold air advection from ECMWF-Interim calculations in the new Fig. 3.

*Lines 332-334: This seems highly speculative.*

We used a careful formulation for this, we do not say that we are talking about facts, but suggest possible reasons. We would not call this speculative.

*Please add DOIs to bibliography*
Done.

*Technical Corrections*

*Line 94: SIC has not been defined. Please define it here and use SIC for the remainder of the study*
Done.

*Line 124: remove "were"*
Done.

*Line 128: change to "December"*
Done.

*Line 130: remove "rather"*

Done.

*Line 137: ECMWF is never defined*

We explained it in introduction section now.

*Line 163: change "today" to "present"*

Done.

*Line 205: please clarify what "Mio." Means*

We changed this according to the Journal style requirements.

*Line 210: no longer need to continue defining SIC, SIE, SIA as they are already defined*

Agreed. We changed this accordingly.

*Line 259: Change "Figure 6" to "Figure 4", and please clarify whether this is sea level pressure (as stated in caption) or surface pressure (as stated in text)*

The figure numbers have changed because of the inclusion of new figures and change of Fig. 4 and we checked the correct usage of the numbers. We corrected surface to sea level pressure in the text.

*Line 269-270: Already defined as the Amundsen Sea Low / ASL, so just use ASL here*

Done.

*Line 275: Please remove the words "masses" and "right"*

We deleted "right", but kept "masses", since "air masses" is a standard meteorological term.

*Line 288: Would say ASL instead of "Pacific low"*

Corrected.

*Line 305: Just put "periods" in parenthesis*

Done.

*Line 307: Please change to Figure 4e*

The figure numbers were changed due to the addition of new figures and are now numbered correctly.